# Retrieving UV-VIS Spectral Single-scattering Albedo of Absorbing Aerosols above Clouds from Synergy of ORACLES Airborne and A-train Sensors

Hiren T Jethva[1,2], Omar Torres[2], Richard Anthony Ferrare[3], Sharon P Burton[3], Anthony L Cook[3], David B Harper[3], Chris A Hostetler[3], Jens Redemann[4], Vinay Kayetha[5], Samuel LeBlanc[6,7], Kristina Pistone[6,7], Logan Mitchell[4], Connor J Flynn[4]

[1]Morgan State University, Baltimore, MD, United States
[2]NASA Goddard Space Flight Center, Greenbelt, MD, United States
[3]NASA Langley Research Center, Hampton, VA, United States
[4]University of Oklahoma, Norman, OK, United States
[5]Science Systems and Applications, Inc., Lanham, MD, United States
[6]Bay Area Environmental Research Institute, Moffett Field, CA, United States
[7]NASA Ames Research Center, Moffett Field, CA, United States

*Correspondence to*: Hiren Jethva (hiren.t.jethva@nasa.gov)

**Abstract**. Inadequate knowledge about the complex microphysical and optical processes of the aerosol-cloud system severely restricts our ability to quantify the resultant impact on climate. Contrary to the negative radiative forcing (cooling) exerted by aerosols in cloud-free skies over dark surfaces, the absorbing aerosols, when lofted over the clouds, can potentially lead to significant warming of the atmosphere. The sign and magnitude of the aerosol radiative forcing over clouds are determined mainly by the amount of aerosol loading, the absorption capacity of aerosols or single-scattering albedo (SSA), and the brightness of the underlying cloud cover. In the satellite-based algorithms that use measurements from passive sensors, the assumption of aerosol SSA is known to be the largest source of uncertainty in quantifying above-cloud aerosol optical depth (ACAOD). In this paper, we introduce a novel synergy algorithm that combines direct airborne measurements of ACAOD and the top-of-atmosphere (TOA) spectral reflectance from OMI and MODIS sensors of NASA's A-train satellites to retrieve 1) SSA of light-absorbing aerosols lofted over the clouds, and 2) aerosol-corrected cloud optical depth (COD). Radiative transfer calculations show a marked sensitivity of the TOA measurements to ACAOD, SSA, and COD, further suggesting that the availability of accurate ACAOD allows retrieval of SSA for above-cloud aerosol scenes using the 'color ratio' algorithm developed for satellite sensors carrying ultraviolet (UV) and visible-near-IR (VNIR) wavelength bands. The proposed algorithm takes advantage of airborne measurements of ACAOD acquired from the High-spectral Resolution Lidar-2 (HSRL-2) and Spectrometer for Sky-Scanning, Sun-Tracking Atmospheric Research (4STAR) Sunphotometer operated during the ORACLES (ObseRvations of Aerosols above CLouds and their intEractionS) field campaign (September 2016, August 2017, and October 2018) over the southeastern Atlantic Ocean, and synergize them with TOA reflectance from OMI and MODIS to derive spectral SSA in the near-UV (354-388 nm) and VIS-near-IR (470-860 nm), respectively. When compared against the ORACLES airborne remote sensing and in situ measurements and the inversion dataset of ground-based AERONET over land, the retrieved spectral SSAs from the satellites, on average, were found to be within an

agreement of ~0.01—the difference well within the uncertainties involved in all these inversion datasets. The retrieved SSA above the clouds at UV-VIS-NIR wavelengths shows a distinct increasing trend from August to October, which is consistent with the ORACLES in situ measurements, AERONET inversions, and previous findings. The sensitivity analysis quantifying theoretical uncertainties in the retrieved SSA shows that errors in the measured ACAOD, aerosol layer height, and the ratio of the imaginary part of the refractive index (spectral dependence) of

aerosols by 20%, 1 km, and 10%, respectively, produce an error in the retrieved SSA at 388 nm (470 nm) by 0.017 (0.015), 0.008 (0.002), and 0.03 (0.005). The development of the proposed aerosol-cloud algorithm implies a possible synergy of CALIOP lidar and OMI-MODIS passive sensors to deduce a global product of ACAOD and SSA. Furthermore, the presented synergy algorithm assumes implications for future missions, such as AOS and EarthCARE. The availability of the intended global dataset can help constrain the climate models with the much-needed

observational estimates of the radiative effects of aerosols in cloudy regions and expand our ability to study aerosol effects on clouds.

## 1   INTRODUCTION

Aerosol-cloud interactions continue to be the largest source of uncertainty in predicting the role of aerosols on climate (IPCC, 2021). One of the main hurdles is the lack of comprehensive knowledge about the complex microphysical and

optical processes of the aerosol-cloud system that govern the resultant impact on the regional and global climate systems. The transoceanic transport of fine-mode aerosol from biomass burning and coarse-mode mineral aerosols from dust storms have been well documented using ground and remote sensing observations (Prospero et al., 2002; Kaufman et al., 2005; Chand et al., 2008; Torres et al., 2012). The long-range transport of aerosols often takes place above low-level clouds. Lofted layers of aerosols above the boundary layer and over the clouds have a longer lifetime

in prevailing wind conditions across the oceans and, therefore, have a higher chance of long-range transport at intercontinental scales. Elevated aerosol layers over clouds are commonly observed during field campaigns (Zuidema et al., 2016; Redemann et al., 2021) as well as from satellites (Chand et al., 2008; Wilcox et al., 2009; Torres et al., 2012; Jethva et al., 2018) over several regions of the world. Unlike the cloud-free conditions in which aerosols generally produce a cooling effect on climate, the presence of elevated layers of absorbing aerosols over clouds can

potentially exert a large positive forcing through enhanced atmospheric heating resulting from aerosol-cloud radiative interactions (Hsu et al., 2003; Keil and Haywood, 2003; Chand et al., 2009; de Graaf et al., 2012; Zhang et al., 2016; Kacenelenbogen et al., 2019). Thus, if not considered in the global estimates, aerosols lofted over the cloud can introduce significant uncertainty in the net aerosol forcing estimates. Even the sign of aerosol forcing will remain unknown until we can produce global, reliable, quantitative observations of aerosol loading above clouds.

Eswaran et al. (2015), using CALIOP, MODIS, and OMI data for the Bay of Bengal region, showed that the critical cloud fraction (CCF)—the cloud fraction at which the sign of aerosol radiative forcing switches from negative (cooling) to positive (warming), is strongly dependent on the single-scattering albedo (SSA) of aerosols above the cloud. A seasonal shift in SSA from about 0.97 to 0.9 reduced the CCF from 0.28 to 0.13. Along the same line, Chand et al. (2009) estimated the critical cloud fraction to be 0.4 for the Southeastern Atlantic Ocean, and further stated that

it is strongly sensitive to the amount of solar radiation absorbed by aerosols, which in turn depends on the columnar

aerosol loading and absorption capacity (SSA) of aerosols and the albedo of the underlying clouds. In addition to their direct effects, absorbing aerosols above the clouds can produce a semi-direct effect on the cloud properties underlying the absorbing aerosol layer (Wilcox, 2012). On the aerosol modeling aspect, different climate models treat aerosol-cloud interaction processes differently, which results in the largest inter-model discrepancies in aerosol forcing assessments, particularly over the absorbing aerosol-cloud overlap regions, i.e., southeastern Atlantic Ocean and Southeast Asia (Schulz et al., 2006).

The magnitude and spatiotemporal extent of these effects are driven by several factors, including the magnitudes of the above-cloud aerosol optical depth (ACAOD), the absorption capacity of aerosols (SSA), and properties of underlying clouds (e.g., cloud optical depth), cloud fraction, and spatiotemporal frequency of absorbing aerosols above the cloud. Satellite observations have unambiguously shown the presence of absorbing aerosols above clouds over several regions of the world on a monthly to seasonal scale (Devsthale and Thomas, 2011; Jethva et al., 2018). These regions include the Southeastern Atlantic Ocean and Southeast Asia, where massive amounts of biomass burning aerosols are often found to overlie low-level stratocumulus cloud deck from June through September and in March and April, respectively; Tropical Atlantic Ocean, where large-scale dust plumes resulting from outbreaks over the Saharan desert overlie oceanic clouds, and Northeast Asia, where dust transported from the Taklimakan and the Gobi deserts possibly mixed with the regional pollution overlays clouds along the eastward transport pathways.

In the past decade, the development of several independent algorithms quantifying aerosol loading above the cloud from satellite-based active and passive sensors has been a breakthrough. These techniques have shown the potential to retrieve ACAOD using measurements from different A-train sensors, including CALIOP/CALIPSO (Hu et al., 2007; Chand et al., 2008), Aura/OMI (Torres et al., 2012), Aqua/MODIS (Jethva et al., 2013; Meyer et al., 2015), Parasol/POLDER (Waquet et al., 2009). It is imperative for the algorithms designed for the passive sensors to assume the microphysical and optical properties of both aerosols and clouds. The theoretical sensitivity analysis documented in these papers shows that the uncertainty in retrieving ACAOD is primarily governed by the optical properties of aerosols assumed in the inversion. In particular, the assumption of aerosol SSA is known to be the largest source of uncertainty in the inversion. For instance, an uncertain value of aerosol SSA by ±0.03 can result in an error of about +50%/-10% in ACAOD at COD of 10 (Torres et al., 2012; Jethva et al., 2013). The errors in the retrieved ACAOD can be even greater, given the larger perturbation in SSA, especially at higher ACAODs coupled with lower CODs. Peers et al. (2015) demonstrated a method based on the polarized and total radiance measurements of POLDER to show that the radiances at 490 nm and 865 nm can be interpreted as a coupled total and absorption ACAOD given the scattering optical depth of aerosol and their size distribution are known.

In this paper, we introduce a novel algorithm that combines the airborne measurements of the above-cloud AOD and satellite measurements of radiances in the near-UV and visible-near-IR regions of the spectrum to retrieve the SSA and aerosol-corrected COD simultaneously for scenes identified with absorbing aerosols overlaying low-level water clouds. This work builds upon the 'color ratio' (CR) technique previously applied to OMI (Torres et al., 2012; Jethva et al., 2018) and MODIS (Jethva et al., 2013, 2016) sensors. Radiative transfer simulations show that *a priori*

information on the ACAOD enables the CR algorithm to retrieve the above-cloud SSA, which otherwise is assumed in the original method.


Section 2 describes the physical basis of the proposed synergy algorithm. The airborne and satellite datasets and their collocation approach are presented in section 3. The optical-microphysical models of aerosols and clouds employed in the algorithm are discussed in section 4. The retrieved spectral single-scattering albedo above clouds and its comparison with equivalent airborne and ground-based measurements/inversions are presented in section 5. The

impact of aerosol absorption on cloud optical depth retrievals is examined in section 6. Expected uncertainties in the retrieved SSA to the input ACAOD and different assumptions made in the algorithm are quantified and discussed in section 7. The concluding remarks, a potential application of the proposed algorithm to the CALIOP-OMI-MODIS synergy, and its implications to the future satellite missions are presented in section 8.

## 2    PHYSICAL BASIS


Light-absorbing carbonaceous particles generated from the biomass burning activities and windblown mineral dust aerosols exhibit a strong wavelength dependence in the UV to visible (VIS) to the shortwave-infrared region (SWIR) (Kirchstetter et al., 2004; Russell et al., 2010). On the other hand, the water clouds exhibit near-neutral wavelength dependence of extinction in this part of the spectrum. Therefore, when absorbing media with strong wavelength-

dependent absorption characteristics lie above the clouds, the net effect observed from space will likely show a stronger spectral signature than what would be expected from just the clouds alone. In other words, this kind of situation produces a strong 'color ratio' effect, which can be seen in the TOA measurements made by satellite sensors such as OMI (Torres et al., 2012), MODIS (Jethva et al., 2013), and OMI-MODIS combined (De Graaf, et al., 2019).

The CR method initially developed and applied to the OMI's near-UV observations retrieves ACAOD and COD

simultaneously under an assumed state of the atmosphere. The inversion method physically relies on an unambiguous color ratio effect, where the TOA reflectance at shorter wavelengths is lower than that at longer wavelengths, caused by absorbing aerosols having strong wavelength-dependent absorption characteristics when overlaying low-level water clouds. The assumptions involved in the retrieval process include aerosol size distribution, aerosol real and imaginary parts of the refractive index, aerosol vertical profile, and optical-microphysical properties of underlying

water clouds. We invoked the radiative transfer (RT) calculations to simulate this effect using a vector discrete ordinate RT model VLIDORT (Spurr, 2006).

Figure 1 shows the simulated reflectance ratio versus single-channel reflectance for the near-UV (354 and 388 nm) and visible wavelength (470 and 860 nm) domains for an atmosphere in which carbonaceous aerosols are assumed to overlay a low-level cloud deck. The particle size distribution (PSD) and spectral properties of aerosols are taken from

the long-term direct and almucantar measurements (Version 3, Level 2) made by the Aerosol Robotic Network (AERONET) sunphotometer at *Mongu* (15.25˚S, 23.15˚E), Zambia in central/southern Africa. The vertical profile of aerosols was assumed to follow a quasi-Gaussian distribution with maximum concentration at 3.0 km with an underlying cloud deck of 500-m depth placed between 1.0 and 1.5 km; both are generally consistent with the vertical

feature mask produced by CALIOP over the Southeastern Atlantic Ocean. Each dot in Figure 1 represents RT simulation for a specific pair of the imaginary part of the refractive index and COD corresponding to a single pair reflectance and color ratio between shorter and longer wavelengths. The presence of absorbing aerosols above the cloud produces a significant reduction in the reflectance and color ratio. For a given AOD, the slope of the solid line depends on both the imaginary index of the lofted aerosol layer above the cloud and the COD of the cloud layer underneath the aerosol layer. Thus, for a given value of AOD, TOA measurements from OMI and MODIS at the respective wavelength pairs can be directly associated with a pair of the imaginary part of the refractive index, which can also be expressed as SSA, and COD. This forms the basis of the proposed above-cloud aerosol retrieval.

## 3    DATASETS

### 3.1    AIRBORNE MEASUREMENTS

ORACLES (ObseRvations of Aerosols above CLouds and their intEractionS) was a five-year NASA EVS-2 (Earth Venture Suborbital-2) airborne campaign conducted over the southeastern Atlantic Ocean and off the western coast of central/southern Africa with three Intensive Observation Periods during the biomass burning seasons of 2016, 2017, and 2018. The investigation was designed to acquire high-quality, in situ, and remote sensing measurements of aerosols and clouds for studying key atmospheric processes that determine the climate impacts of the aerosol-cloud system. A detailed overview of the ORACLES campaign, scientific objectives, and instrumentation is presented in Redemann et al. (2021). Of the various aerosol-cloud measurements acquired during ORACLES, the datasets directly relevant and used in the present work are the direct measurements of spectral aerosol optical depth over clouds from the airborne HSRL-2 lidar and 4STAR sunphotometer. Additionally, we also use the in-situ, inlet-based measurements of spectral SSA acquired from the HiGEAR sensors for the relative comparison against the satellite-based SSA retrievals from OMI and MODIS. In the following sub-sections, we provide a brief description of both datasets.

### 3.1.1    High-spectral Resolution Lidar-2 (HSRL-2)

The HSRL-2 lidar developed at NASA Langley makes detailed vertical measurements of aerosol backscatter and depolarization (355, 532, 1064 nm) and extinction at 355 nm and 532 nm wavelengths via the HSRL technique to characterize aerosols and clouds. The HSRL technique takes advantage of the spectral distribution of the lidar return signal to distinguish aerosol and molecular signals and thereby measure aerosol extinction and backscatter independently, without a priori assumptions on the aerosol type and/or aerosol extinction-to-backscatter ratio. HSRL2 flew onboard NASA's ER-2 aircraft during ORACLES-1 and P3-Orion aircraft during ORACLES-2 and -3 deployments, making a total of 7, 15, and 15 flights, respectively, providing direct measurements of the above-cloud column AOD over the southeastern Atlantic Ocean. Additional details describing the HSRL-2 aerosol measurements are given by Burton et al. (2018), and those acquired during the ORACLES mission are provided by Harshvardhan et al. (2022).

### 3.1.2 Spectrometer for Sky-Scanning, Sun-Tracking Atmospheric Research (4STAR)

The 4STAR is an advanced, next-generation spectrometer developed at the NASA Ames Research Center by the Sunphotometer/Satellite team (Dunagan et al., 2013; Shinozuka et al., 2013; Segal-Rosenheimer et al., 2014; Pistone et al., 2019, LeBlanc et al., 2020), in collaboration with Battelle, Pacific Northwest Division. The 4STAR sensor, in addition to measuring spectral measurements of AOD above the platform in the range 350 nm to 1700 nm, extends the capabilities of the previously developed AATS sensor by adding a sky-scanning mechanism that enables the retrieval of complex refractive index, shape, and aerosol size distribution. 4STAR can also make measurements of column trace gases (e.g., $NO_2$) to enhance the accuracy of aerosol measurements via improved aerosol-gas separation. The 4STAR instrument has been integrated onboard multiple airborne research vessels during different field campaigns, including TCAP during 2012 and 2013, SEAC4RS field experiment based out of Huston, Texas in 2013, Arctic field campaign, ARISE, based out of Fairbanks, Alaska in 2014, NAAMES field mission, based out of St-John's Newfoundland, Canada in 2015, and KORUS-AQ based in South Korea in 2016.

4STAR onboard NASA's P3-Orion aircraft flew over the southeastern Atlantic Ocean during all three ORACLES deployments (September 2016, August 2017, and October 2018) and acquired accurate measurements of spectral AODs of biomass burning aerosols in the region, both in cloud-free and above-cloud aerosols areas. The spectral AODs measured from the direct Sun measurements above the clouds, owing to their higher accuracy, constitute a valuable dataset not just to validate the satellite retrievals of ACAOD but to retrieve SSA when combined with the satellite radiance measurements, as demonstrated in the present work.

In addition to making direct measurements of spectral AODs from UV to near-infrared wavelengths, 4STAR also performed sky scans under favorable flying conditions in either the principal plane or almucantar. The acquired spectral diffuse radiance measurements were processed using a modified version 2 AERONET retrieval algorithm described in Dubovik and King (2000). The AERONET-like inversion algorithm provided aerosol size distributions, refractive indices, SSA, and AAOD, among other parameters. Out of a total of 174 sky scans in ORACLES in 2016, 38 % (66) met the quality control (QC) criteria adapted from the AERONET QC available at https://aeronet.gsfc.nasa.gov/new_web/PDF/AERONETcriteria_final1.pdf, which are also listed in Pistone et al. (2019). Of particular interest to the present work, we used 4STAR sky scan-derived spectral SSAs at wavelengths 400 nm, 500 nm, and 660 nm for the comparison against satellite-derived above-cloud SSA from OMI (388 nm) and MODIS (470 nm). The uncertainties in the 4STAR-retrieved spectral SSAs are discussed later in section 5.5, in which the intercomparison of retrieved SSA from the present work against those derived from other ground and airborne sensors is presented.

### 3.1.3 HiGEAR PSAP and Nephelometer In Situ Observations of Aerosol Scattering and Absorption

During the ORACLES field campaign, aerosol absorption and scattering properties were also measured in situ by PSAP (470, 530, 660 nm) and TSI 3-wavelengths nephelometer (450, 550, and 700 nm) of the Hawaii Group for Environmental Research or HiGEAR, both onboard P3 aircraft. Additionally, two single-wavelength nephelometers

(at 550 nm, Radiance Research) were also operated simultaneously to study the increase in light scattering as a function of relative humidity (RH). The PSAP was not controlled in terms of RH; however, its optical block was heated to approximately 50°C in 2016 and 30°C in 2017 to maintain RH within the nephelometer lower than 40%, thereby reducing the artifacts due to changing RH. We adopted the following steps, as suggested by the HiGEAR team, for selecting the optimum values of in-situ spectral SSA.

1. The PSAP absorption coefficients were averaged using measurements from the front and rear ends of the instrument to increase the number of input data points.

2. Consider measurements when the carbon monoxide (CO) measurements from COMA on board P3 and scattering coefficients from the TSI sensor are greater than the respective median values obtained from all three campaigns combined, i.e., CO of 142 ppbv, TSI scattering coefficient of 45 mm$^{-1}$, 36 mm$^{-1}$, and 27 mm$^{-1}$, at 470 nm, 530 nm, and 660 nm, respectively.

3. Calculate spectral SSA via TSI_Scat/(TSI_Scat + PSAP_Abs) at 470, 530, and 660 nm.

Furthermore, we apply 100-point temporal averaging to the high-frequency SSA measurements to reduce the noise and variability associated with the measurements. The filtered, temporally averaged spectral SSA dataset from the HiGEAR PSAP/Nephelometer is used in this study for the relative comparison against the satellite-based inversion of SSA above the clouds. The aerosol datasets collected by HSRL-2, 4STAR, and HiGEAR teams for all three years of ORACLES field deployments were accessed from the NASA ESPO archive web portal at the URL https://espoarchive.nasa.gov/archive/browse/oracles.

### 3.2    SATELLITE OBSERVATIONS

### 3.2.1    Ozone Monitoring Instrument (OMI)

OMI is a hyperspectral radiometer launched on NASA's Aura satellite in 2004, making measurements of reflected light from the Earth in the spectral range of 270-500 nm to retrieve information on trace gases and aerosols (Torres et al., 2007). With a spectral resolution of 0.45-1.0 nm FWHM and instantaneous field-of-view of 3 km binned to 13x24 km$^2$ spatial resolution at nadir, OMI observes the entire globe daily with an orbital swath width of 2600 km at ground (Levelt et al., 2006). Post-2007, the OMI observations have been affected by a likely external obstruction that perturbs both the measured solar flux and Earth radiance. This obstruction affecting the quality of radiance at all wavelengths for a particular viewing direction is referred to as "row anomaly" (RA) since the viewing geometry is associated with the row numbers on the charge-coupled device detectors. The RA issue was detected for the first time in mid-2007 with a couple of rows, which, over the period of operation, expanded to other rows in 2008 and later. At present, about half of the total 60 rows across the track are identified and flagged as row anomaly affected positions for which no physical retrievals are performed. The details about this issue can be found at http://www.knmi.nl/omi/research/product/rowanomaly-background.php. This has significantly affected the sampling during post-2008 OMI measurements, where about half of the OMI swath is blanketed by row anomaly flags. As a result, the spatial coverage is reduced to about half starting in 2009 compared to earlier OMI measurements.

For the present work, the TOA reflectance measurements at 354 nm and 388 nm wavelengths were extracted from the OMI above-cloud aerosol product—OMACA (Jethva et al., 2018). The proposed inversion algorithm was applied to OMI pixels identified with partially absorbing aerosols lofted above clouds following a scheme adopted in the OMACA algorithm (quality flags '0', '1', and '2').

### 3.2.2    Moderate Resolution Imaging Spectroradiometer (MODIS)

The MODIS sensor launched onboard Terra in 1999 and Aqua in 2002 measures light reflected from Earth in a total of 36 spectral bands spanning the shortwave visible to the infrared region of the spectrum. The TOA reflectance measured in these bands at spatial resolutions of 250 m (band 1-2), 500 m (band 3-7), and 1000 m (band 8-36) are used to retrieve a variety of land, ocean, and atmospheric parameters. In the present application, the MODIS Level 1b spectral reflectance data in the 470 nm (band 3) and 860 nm (band 2) wavelength bands and associated observed geometry at a 1-km spatial resolution were averaged over each successive 5 x 5 pixel grid box ($\sim$5 x 5 km$^2$ at nadir). The averaging was performed over pixels identified as liquid water clouds with confident retrieval of COD>3 extracted from the MODIS 1-km cloud optical depth dataset (MOD/MYD06_L2) (Platnick et al., 2017).

Table 1 summarizes the datasets and characteristics of airborne and satellite sensors employed in this study.

### 3.3    SATELLITE-AIRBORNE SENSORS COLLOCATION

### 3.3.1    Spatiotemporal Matching

Unlike sensors installed on a stationary ground station, the airborne sensor flies on a moving platform both in horizontal and vertical dimensions, requiring continuous tracking for spatially collocating with nearby satellite measurements. During ORACLES-1 operation in late August and most of September of 2016, the HSRL-2 lidar flew onboard the ER-2 platform at a nearly constant altitude of $\sim$20 km, whereas the 4STAR sunphotometer was installed on P3-Orion aircraft sampling the airmass at various altitudes from near-surface to above 6 km along the aircraft trajectory. Both airborne sensors during the ORACLES-2 phase in August and September of 2017 flew on the P3-Orion aircraft, profiling the atmosphere between near-surface to above 6 km.

The OMPIXCOR product of OMI provides ground pixel corner coordinates and associated interception area on the ground, corresponding to 75% of the energy in the along-track field of view (Kurosu and Celarier, 2010). Taking advantage of the OMPIXCOR product, the geolocation coordinates of both HSRL-2 and 4STAR were collocated with OMI by identifying airborne measurements that fall within each OMI pixel polygon defined by the OMPIXCOR four corner coordinates within a time difference of ±1 hour between the satellite and airborne observations. The resulting number of observations of ACAOD from the respective airborne sensors was then averaged and assigned to the corresponding OMI pixels for the above-cloud SSA retrievals. One example of OMI-HSRL-2 collocation for the ER-2 flight operated on Sep 24, 2016, is shown in Supplementary Figure 1. The collocation of MODIS TOA observations with airborne measurements was done by averaging both datasets in each regular grid box of size 0.25° x 0.25° within the time difference of ±1 hour.

### 3.3.2 Adjustment in 4STAR ACAOD

Spectral AOD measurements from the 4STAR onboard the P3-Orion aircraft correspond to an atmospheric column above the aircraft altitude. The P3 aircraft, during all three years of the ORACLES campaign, profiled the atmosphere between the surface to ~6 km and made a limited number of measurements of AOD when the aircraft flew below the aerosol layer and just above the cloud top. Full utilization of the 4STAR dataset requires an adjustment in the altitudinal measurements of AOD that extrapolates the AOD measured at different altitudes to the cloud top. Such adjustments were performed on each valid measurement of 4STAR based on a quadratic polynomial between altitude and AOD calculated using good-quality observations made during each flight. Figure 3 displays the altitude vs. AOD polynomial of the three relevant wavelengths (388 nm, 470 nm, 501 nm) for a few select flights operated during the three years of ORACLES. Similar plots for all P3 fights with 4STAR AOD profiles derived for all three years are included in supplementary figures 2, 3, and 4. First, the quadratic polynomial relating altitude and AOD for each flight was derived using all good quality data (qual_flag=0), including those measurements that are flagged as AOD above the cloud (flag_acaod=1). In the next step, the outlier data points that deviate from the polynomial fit by greater than 15% in AOD were rejected. The altitude vs. AOD polynomials were re-calculated with the remaining smoothly varying data points and are used to adjust the altitude-dependent AOD to the cloud-top as follows.

In the first step, AODs recorded at GPS altitude ($AOD_{GPSAlt}$) and cloud-top altitude ($AOD_{CldTopAlt}$) are calculated from the quadratic polynomials as,

$$AOD_{GPSAlt} = C0 + C1 * GPSAlt + C2 * GPSAlt^2$$

$$AOD_{CldTopAlt} = C0 + C1 * CldTopAlt + C2 * CldTopAlt^2$$

Second, the difference between the AODs measured by 4STAR ($AOD_{4STAR}$) and that calculated from the polynomial is calculated as,

$$\Delta AOD = AOD_{4STAR} - AOD_{GPSAlt}$$

In the last step, the adjusted AOD representing columnar aerosol amounts above the cloud-top ($AOD_{AdjtoCldTop}$) is calculated by adding the difference in AOD ($\Delta AOD$) calculated in the previous step to the AOD calculated at cloud-top ($AOD_{CldTopAlt}$) as,

$$AOD_{AdjtoCldTop} = AOD_{CldTopAlt} + \Delta AOD$$

The information on cloud-top altitude for each collocated 4STAR measurement was obtained from the collocated MODIS standard MOD/MYD06 cloud product. The altitude-dependent AOD measurements adjusted to the cloud-top following the above-described procedure were used as input to the proposed SSA inversion algorithm.

We use all good-quality AOD measurements collected during a particular P3 flight to derive the altitude-AOD polynomial specific to that flight, and therefore, it stands as an average representation of the AOD profile over the entire flight track. The actual AOD profile below the aircraft level for an instance of 4STAR measurements might differ from the average vertical AOD profile for the entire flight. Any deviation in the actual vertical AOD profile,

therefore, may result in erroneous above-cloud columnar AOD calculated from the procedure described above. In section 7, we discuss the resultant error in the retrieved above-cloud aerosol SSA due to various input factors, including input ACAODs. It is to be noted here that no such adjustments were made to the collocated 4STAR ACAOD measurements when the P3 aircraft was flying just above the cloud top and below the aerosol layer, as appropriately flagged in the 4STAR dataset.

## 4    RETRIEVAL ALGORITHM

The physical basis of the proposed retrieval method discussed in the previous section relies on specific assumptions about the optical and microphysical properties of aerosols and clouds. In this section, we describe each of these assumptions as follows.

### 4.1    AEROSOL MODEL PROPERTIES

The aerosol optical and microphysical models required to generate look-up-tables (LUTs) of spectral TOA reflectance were derived from the multi-year statistics of the AERONET cloud-free, Level 2, version 3 direct measurements and inversions carried out at *Mongu* (15˚S, 23˚E), Zambia, in southern Africa. Multi-year aerosol measurements taken from July through September, when biomass burning activities are widespread over central/southern Africa (Eck et al., 2013), are considered. The particle size distribution is assumed to follow a bi-modal lognormal distribution with the mean and standard deviation of radius for both fine and coarse modes given by the long-term inversion dataset of AERONET (Holben et al., 1998; Dubovik et al., 2002). Similarly, the real part of the refractive index and the Extinction Ångström Exponent in the visible and near-UV spectral domains were also assumed based on the AERONET inversion and direct measurements datasets, respectively. The LUT nodes in the imaginary index at 388 nm and 470 nm encompass a wide range from very absorbing to fully scattering aerosols. The spectral dependence of the imaginary part of the refractive index in the visible-near-IR wavelength range (470-860 nm) is described by the AERONET dataset, whereas that for the near-UV is taken from the OMI/OMAERUV carbonaceous aerosol model (Jethva and Torres, 2011).

To represent the aerosol profile, we take advantage of the detailed vertical measurements of particulate extinction measured by the HSRL-2 lidar. The extinction-weighted aerosol layer height (ALH) for each collocated set of measurements of HSRL-2 was calculated by weighing the altitude grids with their corresponding extinction measurements. The aerosol vertical profile is assumed to follow a quasi-Gaussian distribution around the mean ALH given by the collocated HSRL-2 extinction measurements. In September 2016, HSRL-2 flew on the ER-2 platform at an altitude of ~20 km, measuring the detailed vertical distribution of aerosol extinction in the entire atmospheric column below 20 km and above the clouds in the lower troposphere. The overall extinction-weighted mean (standard deviation) aerosol layer height for a total of 7 ER-2/HSRL-2 flights was calculated as 3.49 (±0.56) km. In August 2017 and October 2018, HSRL-2 was mounted on P3-Orion aircraft and measured the aerosol profiles below the aircraft level and above the cloud top, resulting in an overall mean aerosol layer height of 2.19 (±0.51) and 2.26 (±0.71), respectively. The overall lower aerosol layer height calculated during the latter two years could be likely due to the missing aerosols above 6 km not measured by HSRL-2.

Such detailed vertical information on aerosol profiles is hard to extract from the altitude-dependent 4STAR AOD dataset or is limited to the times of aircraft profiles. Based on the aerosol layer height statistics obtained from HSRL-2 altitude-resolved measurements, therefore, the retrievals of SSA from the satellite-4STAR collocation data were performed assuming a nominal ALH of 3.0 km above mean sea level. The ocean surface is assumed to be Lambertian with fixed values of surface albedo of 0.05 and 0.03 in the near-UV (354 and 388 nm) and visible-near-IR region

(470-860 nm).

**4.2     CLOUD MODEL PROPERTIES**

The proposed method to retrieve aerosol SSA is designed to perform retrievals over low-level liquid water clouds. Cloud droplet size distribution follows the modified Gamma distribution (Deirmendjian, 1969) formalized as,

$$n(r) = ar^\alpha \exp\left(-\beta r^\gamma\right)$$

where, the constants $\alpha$, $\beta$, and $\gamma$ parameters are assumed as 15.0, 1.5, and 1.0, respectively. The cloud effective radius (CRE), when calculated from these parameters, turns out to be 12.0 μm. The choice of the cloud effective radius value is based on the Aqua/MODIS standard cloud retrievals over the Southeastern Atlantic Ocean. The supplementary Figure 5 shows the histogram of CRE data derived from the standard Aqua/MODIS MYD06 cloud product over the Southeastern Atlantic Ocean from August through October 2016 to 2018. The CRE histograms for the months of

August to October reveal a distinct peak around 11-12 μm with a wide distribution ranging from 6 μm to 26 μm, albeit 70%-80% of the data points are found to be within 6-18 μm. Meyer et al. (2015) have shown that the presence of partially absorbing carbonaceous smoke aerosols over clouds in this region has only a marginal impact of ~%2 increase in the retrieved CRE from MODIS. The real part of the refractive indices at 354-388 nm and 470-860 nm were obtained from Hale and Querry (1973). A homogeneous cloud layer with a geometric thickness of 500 meters is assumed to be

located beneath the aerosol layer and between 850 hPa and 900 hPa pressure levels or approximately 1.5 km and 1.0 km altitudes. The assumed cloud top and bottom altitudes are in close agreement with CALIOP detection of cloud top (~1.2-1.4 km) and bottom (~0.5-0.7 km) altitudes over the Southeastern Atlantic Ocean during August through October. The TOA radiances at the near-UV and visible-near-IR wavelengths were calculated for a total of ten nodes in COD, i.e., 0, 2, 5, 7, 10, 15, 20, 30, 40, and 50.

**4.3     RADIATIVE TRANSFER CALCULATIONS**

The proposed algorithm relies on a look-up-table search to find a pair of above-cloud aerosol SSA and underlying COD that explains the TOA reflectance observations from OMI and MODIS independently. We employ the VLIDORT code (Spurr, 2006) to generate the required LUTs for the carbonaceous aerosol model. The code offers the full linearization ability with the outputs of the Stokes vector for arbitrary viewing geometry and optical depth of

aerosols and clouds. The VLIDORT model results were compared against the published literature and the RT simulations from benchmark in-house RT codes used within the trace gases and aerosol groups at NASA Goddard. The VLIDORT code was further upgraded for the joint aerosol-cloud simulations and used to create look-up tables

for the above-cloud AOD retrieval from Aura/OMI, DSCVOR-EPIC, and S5p/TROPOMI sensors. The validation of the ACAOD product from these sensors against the ORACLES airborne direct measurements revealed a satisfactory agreement within the expected uncertainties (Jethva et al., 2018; Ahn et al., 2021; Torres et al., 2020), thereby establishing the reliability of the VLIDORT RT code in the present application. The LUTs are referenced to several nodes in ACAOD, COD, the imaginary part of the refractive index, and Sun-satellite viewing geometries listed in Table 2. The observed spectral reflectance from OMI and MODIS are fitted in the 2D retrieval domain calculated for a measured value of ACAOD given by airborne sensors, such as shown in Figure 1, to derive above-cloud SSA and aerosol-corrected COD simultaneously. A simplified flow chart of the proposed airborne-satellite synergy algorithm is shown in Figure 4.

## 5    RESULTS

### 5.1    ACAOD MEASUREMENTS FROM HSRL-2 AND 4STAR

Figure 2 shows the flight tracks of ER-2 and P3-B color-coded with measurements of ACAOD (500 nm) from HSRL-2 and 4STAR sensors operated during August and September of 2016 (a, c) and 2017 (b, d). Each sub-figure also shows the corresponding histogram of ACAOD. Both sensors measured a range of ACAOD (0.0-1.0), with the majority of observations falling between the range of 0.2-0.4. The 4STAR aerosol dataset provides a flag for measurements taken when the aircraft flew above the cloud deck but below the aerosol layer. These measurements are flagged as total column ACAOD, which are shown in the figure. In the proposed inversion algorithm, we use all 4STAR ACAOD measurements taken between aircraft altitudes 1 km and 4 km and adjusted to the cloud-top following the collocation procedure described in section 3.3. The HSRL-2 lidar measured ACAOD from an aircraft altitude of ~20 km from ER-2 in 2016, whereas it flew onboard P3-Orion in August 2017 and October 2018 and measured ACAOD when the aircraft was flying at an altitude of about 5.6 km (averaged over all flights in 2017 and 2018). It is to be noted here that any aerosol layers located above the P3-Orion aircraft altitude are not measured by the HSRL-2, and therefore, will be missed in the ACAOD measurements, leading to underestimation in columnar aerosol loading in such cases.

### 5.2    NEAR-UV SSA RETRIEVALS FROM OMI-ORACLES SYNERGY

The spatial distribution of retrieved above-cloud aerosol SSA at 388 nm wavelength along the flight tracks from OMI-ORACLES synergy is shown in Figure 5. The results derived from OMI-HSRL-2 (OMI-4STAR) are shown in the left (right) panels. Each subplot also displays the corresponding histogram of retrieved SSA at 354 nm (blue) and 388 nm (red). The retrievals of SSA shown in the figure correspond to the collocated airborne ACAOD (500 nm) > 0.2 measured from HSRL-2 and 4STAR. Among all four synergies, the OMI-HSRL-2 yields the maximum number of matchups (N=140) during the 2016 campaign due to broader spatial coverage of ACAOD measured from HSRL-2 than was available from 4STAR or from HSRL-2 on P3 in 2017 and 2018. Note that about half of the OMI swath was affected by the row anomaly during the ORACLES operation period, resulting in fewer matchups between valid OMI and airborne observations. The retrieved SSA at UV wavelengths for the majority of the matchups is found in the

range of 0.88-0.91, with an ACAOD-weighted value of 0.88 (0.89) at 354 nm (388 nm). A few matchups obtained from the August 2017 and October 2018 campaigns were due to the limited availability of HSRL-2 ACAOD measurements from P3 aircraft.

The OMI-4STAR synergy yielded a total of 52 matchups in September 2016, with the retrieved SSA at both UV wavelengths to be 0.91-0.92, larger by 0.03 than those obtained from the OMI-HSRL-2 combinations. For the latter two years of ORACLES deployments, i.e., August 2017 and October 2018, OMI-4STAR provides significantly greater matchups than HSRL-2-OMI due to more availability of altitude-dependent AOD measurements, which are adjusted to the cloud top following the procedure described in section 3.3.2. The retrieved above-cloud SSAs at 354 nm and 388 nm were in the range of 0.84-0.92, with the histograms peaking at 0.88 and 0.89 at the respective wavelengths. For the October 2018 deployment, a total of 19 matchups of OMI-4STAR observations yielded SSA distribution, peaking at 0.92 and 0.93 at UV wavelengths.

### 5.3    VIS SSA RETRIEVALS FROM MODIS-ORACLES SYNERGY

Figure 6 shows maps of retrieved above-cloud aerosol SSA at 470 nm wavelength from MODIS-ORACLES synergy. A noticeable feature of SSA retrievals in the visible channel is a significant increase in satellite-airborne matchups relative to OMI near-UV retrievals owing to the availability of original higher spatial resolution measurements at 1 km re-gridded to 5 x 5 km box at the nadir. Again, the MODIS-HSRL-2 collocation provided maximum retrieval matchups (N=298), followed by those with 4STAR (N=154) for the August 2016 observations. The histogram of the retrieved SSA at 470 nm (blue) from both HSRL-2 and 4STAR synergies shows a range of values between 0.84-0.96, peaking around 0.89 during ORACLES-1 operation (September 2016). For the 860 nm wavelength, the retrieved SSA peaked around 0.83-0.85. For the ORACLES-2 operation (August 2017), the total number of airborne-satellite matchups was significantly lower, yielding ACAOD-weighted SSAs of ~0.86 and ~0.81, at both wavelengths, respectively, which are lower than the SSA values derived from ORACLES-1 observations. The airborne-MODIS synergy resulted in the lowest number of matchups for ORACLES-3 October 2018 deployment due to a limited number of ACAOD (500 nm) measurements greater than 0.2. Both HSRL-2 and 4STAR synergies with MODIS retrieve above-cloud SSAs of 0.90 and 0.85 at 470 nm and 860 nm, respectively, which represent the relatively largest values of SSA among all three years of ORACLES measurements.

### 5.4    DEPENDENCY OF RETRIEVED SSA ON ACAOD

The proposed satellite-airborne synergy algorithm used all airborne ACAOD observations > 0.1 at 500 nm collocated with OMI and MODIS sensors. It is expected that the inversion of SSA and cloud optical depth should be robust at higher values of ACAOD owing to greater sensitivity to TOA reflectance and color ratio, hence better resolved retrieval domain space, as shown in Figure 1. To assess the dependency of the retrieved SSA to the input ACAOD, the retrievals are binned as a function of ACAOD and plotted as a box and whisker plot in Figure 7. The top (bottom) panel shows OMI-retrieved SSA at 388 nm (MODIS-retrieved SSA at 470 nm) as a function of collocated airborne ACAOD measured from HSRL-2 lidar and 4STAR sun photometer combined during all three years of ORACLES. The mean and median values of SSA for each AOD bin size of 0.1 are shown as dots and solid lines, whereas the

shaded box contains 50% percentile of data. A noticeable feature in both plots is the tendency of the synergy algorithm to retrieve relatively lower values of SSA at the near-UV and VIS wavelengths for the AOD bin 0.1-0.2. At such lower values of ACAOD, the 2D retrieval domain (not shown in Figure 1) collapses to a very narrow region, i.e., the sensitivity to TOA reflectance and color ratio diminishes significantly. Given the assumptions in the algorithm and their associated uncertainties, it is practically harder to retrieve SSA reliably at lower ACAOD. However, at moderate to larger values of ACAOD, the retrieved SSAs at both wavelengths are largely stable and do not show a well-defined behavior with ACAOD. The variations in the retrieved SSA, shown as shaded boxes and whiskers, therefore, largely attributed to the actual variations in the aerosol absorption within the spatial and temporal domain of ORACLES. This analysis has suggested that the retrieved SSA at much lower ACAOD (~0.1-0.2) at the mid-visible wavelength could be biased low, whereas retrievals are found to be stable at moderate to higher ACAODs (>0.2).

## 5.5    COMPARATIVE EVALUATION OF MULTI-SENSOR SSA RETRIEVALS

The satellite-based SSA retrievals in clear-sky conditions as well as above the cloud proposed in this work are hard to validate due to the scarcity of equivalent columnar, in situ measurements of aerosol absorption. Such validation efforts require well-coordinated airborne, full atmospheric column measurements of aerosol absorption properties aligned with satellite overpass time. To evaluate the relative agreement and differences in the retrieved above-cloud aerosol SSA from the present synergy, we used the following three independent datasets of SSA acquired over the southeastern Atlantic Ocean during the ORACLES period: 1) the aerosol SSA dataset derived from absorption and scattering properties measured in situ onboard P3 by PSAP (470, 530, 660 nm) and TSI 3-wavelengths nephelometer (450, 550, and 700 nm) of HiGEAR team, 2) ground-based inversion of spectral SSA from AERONET, and 3) UV-VIS spectral SSA dataset retrieved from the synergy of AERONET-OMI-MODIS at *Mongu* site from the work of Kayetha et al. (2022).

### 5.5.1    Comparison with In-situ Measurements and AERONET Ground Inversion

Figure 8 shows a composite comparison chart of the retrieved spectral SSA above the cloud from the present work and those obtained from the above-mentioned three spectral SSA datasets for the ORACLES-1 (top) and -2 (bottom) campaigns. The above-cloud SSA retrievals from OMI (354 nm, 388 nm) and MODIS (470 nm, 860 nm) are shown as a standard box and whisker (BW) plot format, where the mean and median are presented as filled circles and horizontal lines; shaded boxes (light blue for OMI and orange for MODIS) represent data points contained within 25-75 percentiles; and thin vertical lines as 1.5 times interquartile range (75 minus 25 percentile). The corresponding SSA retrievals shown at the 660 nm wavelength were derived by linearly interpolating SSA between 470 nm and 860 nm. The in situ spectral SSA dataset derived from PSAP+nephelometer measurements for the respective two years of the ORACLES campaign is shown as red-outlined BW at corresponding measurement wavelengths of 470 nm, 530 nm, and 660 nm. Note that these datasets were not spatially and/or temporally collocated. Instead, the respective datasets collected during the three years of ORACLES were aggregated separately to derive statistics shown in Figure 8.

For the ORACLES operation in September 2016, the mean value of above-cloud aerosol SSA at visible wavelengths from MODIS is found to be in close agreement with that measured from the PSAP+nephelometer. However, the MODIS SSAs contained within the 25-75 percentile range were overall higher by ~ 0.01. The AERONET inversion

of spectral SSAs at a couple of inland ground stations were also in agreement with the 25-75 percentile range of the MODIS SSAs retrieved above the clouds at three wavelengths of 470 nm, 670 nm, and 860 nm. A similar comparison of the retrieved SSA above the clouds for the near-UV wavelengths was not possible due to the non-availability of in situ measurements of SSA and inversion dataset from AERONET. However, the retrieval dataset of UV-VIS spectral SSAs derived from the AERONET-OMI-MODIS synergy at the AERONET site of *Mongu* is used for the relative comparison, which shows an overall 0.01 overestimation in the OMI-retrieved SSA and about 0.01-00.2 overestimation in the MODIS-retrieved SSA at the visible wavelengths.

Similar comparison results were obtained for the ORACLES-2 operation in August 2017 (Figure 8, b), where the agreement and differences noticed for the ORACLES-1 deployment also apply here, albeit with two noticeable changes: 1) the absolute values of spectral SSA from all these different datasets were lower by ~0.02 compared to the September 2016 results, and 2) the spread in the in situ SSA measurements at all three visible wavelengths was relatively smaller. The multi-sensor SSA retrieval comparison for ORACLES-3 deployment in October 2018 shown in Figure 8 (c) reveals satellite retrievals of SSA being higher by 0.01-0.02 at the visible wavelengths and by ~0.03 at the near-UV wavelengths. The AEROENT-retrieved SSAs over the continent were within the 25-75 percentile range of satellite retrievals at the VIS-NIR wavelengths. Overall, the SSA values retrieved from different sensors and methods for October 2018 were higher by ~0.02 compared to those observed in September 2016.

We noticed two salient features in the presented multi-sensor, multi-method inversion datasets of SSA. First, the proposed satellite retrievals of above-cloud SSAs agree with the in situ and ground-based remote sensing inversion within the difference of 0.01 to 0.02. Second, all these distinct SSA retrievals, despite their different types of measurements employing different inversion techniques, capture an intra-seasonal change in the retrieved SSA over the Southeastern Atlantic Ocean that are in sync with the findings of previous studies. Eck et al. (2013) have shown, using the AERONET ground inversion SSA dataset, a significant increase in SSA (440 nm) from ~0.84 in July to ~0.93 in November in southern Africa during biomass burning season. A significant downward seasonal trend in imaginary refractive index, or increased SSA, suggested a gradual decrease in black carbon content in the aerosol composition along the progression of the burning season. The study also confirmed, using the SSA retrievals at 388 nm from OMI, that the seasonal trend in aerosol absorption was not just specific to the AERONET sites, but also observed regionwide.

A similar increase in UV-VIS SSA retrieved above clouds from OMI and MODIS during the three deployment years of the ORACLES instrumentations, capturing the seasonal behavior of the atmospheric composition, is in-line and consistent with the findings of Eck et al. (2013). Furthermore, our results reveal that the seasonal change in aerosol absorption noted over the continent is also observed over the adjacent Southeastern Atlantic Ocean.

### 5.5.2    Comparison with 4STAR Remote Sensing Inversion of SSA
The retrieved above-cloud aerosol SSA from OMI and MODIS are also compared against those derived from sky scan observations made by the 4STAR sunphotometer. During the ORACLES-1 September 2016 operation, 4STAR on P3 Orion aircraft made the measurements of diffuse sky radiation at five wavelengths, i.e., 400 nm, 500 nm, 675 nm, 870 nm, and 995 nm. Using an AERONET-like inversion procedure, these sky radiances are inverted to retrieve particle

size distribution and the real/imaginary part of the refractive index. The errors in the 4STAR-retrieved SSAs are mainly driven by the uncertainties in the input AOD and sky radiances. The uncertainties in the measured AOD are wavelength-dependent and also vary with the time of measurements and solar zenith angle. These values were typically between 0.01 and 0.02, ranging from a low of 0.008 to a high of 0.037 in an extreme case. On the other hand, uncertainties in radiance measurements are quantified between 1.0 % and 1.2 % for the 470-995 nm spectral range, which is wavelength dependent but constant for the entire 2016 campaign (Pistone et al., 2019).

Figure 9 compares spatiotemporally collocated OMI- and 4STAR-retrieved SSA for the ORACLES-1 September 2016 campaign. OMI retrievals of above-cloud SSA were spatially collocated within 1° square box centered at the 4STAR/P3-B geolocation for the days when 4STAR made successful sky scans followed by SSA inversions, as indicated in the legends (bottom-left) with different colors. Each circle in the figure represents a single matchup between the two sensors, where the size of the circle denotes the range of coincident AOD [500 nm] measured by 4STAR. The OMI-4STAR SSA comparison was performed for the two sets of OMI retrievals, i.e., one derived from using the original aerosol model listed in Table 2 and one with the modified aerosol model (see the following paragraph). The OMI-4STAR matchups are found to be in good agreement within ±0.03 difference for the P3 flights operated on September 12 and 14, for which the corresponding AODs were in the range 0.2-0.4. For other flights, OMI SSAs were biased high by ~0.05 relative to the 4STAR inversions. Overall, the comparison yields a root-mean-square-difference (RMSD) of 0.054 and bias of 0.052 with close to 55% (5%) matchups agreeing within ±0.05 (±0.03) difference.

One of the assumptions to which the TOA radiances in the near-UV region are sensitive is the assumed spectral dependence of aerosol absorption or Absorption Ångström Exponent (AAE) both in cloud-free (Jethva and Torres, 2011) and above-cloud aerosols scenes (Jethva et al., 2018). The optical properties assumed in the present near-UV aerosol model are adopted from the standard OMI-OMAERUV cloud-free aerosol algorithm, which has provided a good agreement in both AOD and SSA retrievals with those of ground-based AERONET globally (Ahn et al., 2014; Jethva et al., 2014). However, a departure of the assumed spectral properties from those of actual aerosols may introduce errors in the retrieved above-cloud SSA. For the sake of sensitivity analysis, we modified the original near-UV aerosol model by reducing the relative spectral dependence of the imaginary part of the refractive index from 20% (the original assumption) to 10% between 354-388 nm, resulting in lower AAE of ~1.73-1.87 compared to AAE of ~2.45-2.60 assumed in the original aerosol model. This is achieved by lowering the imaginary part of the refractive index at 354 nm, keeping it the same for the 388 nm wavelength. As shown in Figure 9 (b), the above-cloud SSA retrieved using the modified aerosol model shows relative improvements with reduced RMSD (0.03) and an increased number of matchups within 0.03 (60%) and 0.05 (95%) error limits. However, a positive relative bias of 0.026 remains.

A similar comparison of the retrieved SSA at 470 nm and 670 nm from MODIS against those of 4STAR inversions is presented in Figure 10. During ORACLES-1 deployment in September 2016, 4STAR took sky measurements and retrieved SSA at 400 nm, 500 nm, 675 nm, 870 nm, and 995 nm wavelengths. To facilitate a direct comparison with satellite retrieval, the 4STAR SSAs were linearly interpolated to the 470 nm wavelength. MODIS 470-nm SSAs compare relatively better with RMSD and bias of 0.034 and 0.024, respectively, where about 52% and 73% of the

matchups are found within 0.03 and 0.05 differences, respectively. Both inversions show relatively greater absorption (lower SSA) at 670 nm, generally consistent for the fine mode particles; however, yielding slightly larger RMSD and bias compared to those at 470 nm.

For the ORACLES-2 deployment in August 2017, 4STAR inversion dataset reports SSA at four wavelengths, excluding the shortest 400 nm wavelength due to contamination issues. Therefore, the 4STAR SSAs linearly extrapolated to 470 nm based on the 500-675 nm SSAs. The comparison in Figure 10 (b) shows greater variability in the 4STAR SSAs in the range 0.80-0.95, where 34% (22%) and 53% (42%) of MODIS-retrieved 470 nm (670 nm) SSAs are in agreement within 0.03 and 0.05 limits, respectively. No meaningful relation between the disagreement in SSAs and measured AODs was noticed as several airborne-satellite matchups having both low to moderate and higher aerosol loadings were found to agree well within ±0.03 differences. SSA matchups having disagreements larger than 0.05 are those with 4STAR SSAs in the range 0.86-0.94, but MODIS-retrieved SSAs are still lower than 0.86 at both 470 nm and 670 nm wavelengths. Several factors, including spatiotemporal mismatch, columnar (satellite) vs. above-aircraft (4STAR) representation of SSAs, and inherent uncertainties in both inversion techniques, could be attributed to the observed discrepancies, which need further investigation.

## 6  IMPACT OF AEROSOL ATTENUATION ON CLOUD OPTICAL DEPTH RETRIEVALS

The lofted layers of absorbing aerosols over the clouds attenuate light reflected by the cloud top through scattering and absorption. This effect reduces cloud-reflected upwelling UV (Torres et al., 2012) and VIS-NIR radiation (Jethva et al., 2013; Meyer et al., 2015), reaching the TOA and measured by the satellite sensors. Therefore, retrievals of COD derived from passive sensors such as OMI and MODIS in the UV to VNIR spectrum are expected to be biased low if absorbing aerosols are not accounted for in the inversion. The magnitudes of bias in the apparent COD depend on the strength of aerosol absorption and backscattering as well as on the actual value of COD (Haywood et al., 2004; Meyer et al., 2015; Jethva et al., 2018). The proposed satellite-airborne synergy algorithm retrieves two sets of COD, one corrected for the presence of absorbing aerosols overlying the cloud deck and one retrieved assuming no aerosols above the cloud, which is termed as the apparent COD. Comparing these two sets of CODs is worthwhile to estimate the bias in COD retrievals due to the presence of aerosols above clouds.

Figure 11 (a) shows a comparison of the aerosol-corrected COD (y-axis) to the apparent COD (x-axis) retrieved from both OMI and MODIS observations at 388 nm (red) and 860 nm (blue), respectively. The synergy retrievals from HSRL-2 and 4STAR ACAOD (>0.2 at 500 nm) datasets acquired during all three ORACLES campaigns were used here. The aerosol-corrected CODs are found to be higher compared to apparent CODs (not corrected for aerosols) by an average (standard deviation) value of 26% (±20%) and 9% (±7%) at 388 nm and 860 nm, respectively. In other words, if not accounted for the presence of absorbing aerosols above the clouds, the retrieved COD turns out to be about -18% (±14%) and -8% (±6%) biased low compared to the aerosol-corrected CODs at UV and NIR wavelengths, respectively. The linear regression fits show positive slopes of 1.56 and 1.21 at 388 nm and 470 nm, respectively, with a negative intercept of ~1-1.5, which points to the inherent uncertainties involved in the proposed inversion technique at lower COD values.  A similar comparison displayed inside Figure 11 (a) shows that the aerosol-corrected cloud optical depth retrieved from the present work is higher overall by 16% compared to that obtained from the OMI-

OMACA standard product. The difference between the two sets of retrievals reflects the difference in the cloud effective radius assumed in both inversion algorithms, i.e., 12 μm in the present work versus 6 μm in the OMI-OMACA product. This result is important feedback to the OMACA product to consider revising the cloud model, particularly, the cloud effective radius assumption, for the southeastern Atlantic Ocean.

Figure 11 (b) and (c) further parameterize the same data group and present the % difference in COD, i.e., aerosol-corrected minus apparent (non-corrected), as a function of above-cloud aerosol absorption AOD (AAOD) at 388 nm of OMI and 470 nm of MODIS, respectively. Each colored box represents 25 to 75-percentile data grouped in an AAOD bin size of 0.01. Different colors denote data groups corresponding to different ranges of CODs. At both UV and NIR wavelengths, the magnitudes of % difference in COD have strongly correlated with and turn out to be a bifunction of AAOD as well as the actual values of CODs. The differences between the aerosol-corrected and non-corrected COD not only increase with increasing aerosol absorption above clouds (AAOD) but are also enhanced by larger COD underneath the aerosol layer. For instance, at a given value of AAOD, the differences between the aerosol-corrected and non-corrected CODs are found to be larger at higher CODs. These results are significant and further signify the importance of aerosol absorption above clouds in the UV to VIS-NIR spectral region in at least two ways. First, aerosol absorption above clouds, if not accounted for in the remote sensing inversion, can potentially introduce a negative bias in the retrieved cloud optical depth retrieval, whose magnitude depends on the strength of aerosol absorption (AAOD) and cloud brightness (COD) underneath the aerosol layer. Second, an accurate estimate of the radiative effects of aerosols above the cloud requires the true values of COD along with AAOD. Both these research avenues are currently out of the scope of the present analysis; however, they should be further investigated in depth in future studies.

## 7    POSSIBLE SOURCES OF UNCERTAINTIES

The sensitivity analysis entirely based on radiative transfer simulations is a standard, well-accepted approach for quantifying the theoretical uncertainties in satellite-based inversions. On the other hand, quantifying the actual uncertainties involved in various assumptions made in the algorithm is a daunting task since the information on the true state of aerosol and cloud parameters is often unavailable. Therefore, we adopted another approach in which the radiative transfer simulations, or aerosol-cloud look-up tables in the present work, were first carried out by considering uncertainties in different algorithmic assumptions individually. In the next step, the revised aerosol-cloud look-up tables were used in the inversion algorithm and applied to the actual airborne-satellite measurements. The resultant changes in the retrievals were then interpreted as the expected uncertainties in the retrieved spectral above-cloud aerosol SSA caused by realistic uncertainty in each algorithmic assumption separately. The suggested approach integrates different sets of radiative transfer simulations and actual observations to derive more realistic estimates of the errors in the derived aerosol SSA retrievals.

### 7.1    ERROR IN SSA DUE TO UNCERTAIN ACAOD

When synergized with satellite observations, the direct measurements of ACAOD from airborne sensors allow for retrieving above-cloud aerosol SSA (see Figure 1). The accuracy of retrieved SSA, therefore, is primarily governed

by, in addition to the other algorithmic assumptions, the accuracy of measured ACAOD and its spatiotemporal collocation with satellite observations used as input to the inversion algorithm. Additionally, the imperfectness in collocating airborne and satellite datasets may introduce a mismatch between observed AOD conditions and corresponding reflectance from the satellite. For instance, AODs from airborne sensors are columnar point measurements, whereas satellite observations correspond to the pixel area projected on the ground. Furthermore, the limited availability of airborne measurements in the given pixel polygons (OMPIXCOR coordinates product for OMI and 0.25 deg. grid box for MODIS) may not fully represent the atmospheric conditions observed by the satellite sensor in the pixel area or chosen spatial windows.

To quantify the error in the retrieved SSA due to the uncertainty in the input ACAOD, we adopted a perturbation-based approach, in which the observed ACAOD was perturbed in both directions, i.e., underestimation and overestimation. The resultant retrievals of SSA were compared against those obtained from the original set of input ACAODs. The direct measurements of columnar ACAOD above the cloud deck do not require adjusting the measurement to the cloud-top, such as performed on the 4STAR ACAOD data, thereby reducing the uncertainty in the input ACAOD to the synergy algorithm. For this reason, we selected the collocated HSRL-2 and OMI/MODIS datasets of the ORACLES-1 campaign to perform the SSA uncertainty analysis. The retrievals of above-cloud SSA were performed with the perturbed airborne ACAODs in the range -40% to +40% in step of 10%. The retrieved SSA in each step of the perturbation was compared against the original SSA derived from the unperturbed ACAODs to calculate the corresponding errors in the SSA.

The error matrix in the derived above-cloud SSA from the described procedure is shown in Figure 12 as well as tabulated in Table 3 for the 388 nm and 470 nm wavelengths. For the OMI-based near-UV SSA retrievals, an error of -40% (underestimation) in ACAOD results in an absolute error of -0.054 in the retrieved SSA, the magnitude of which decreases to about -0.018 and -0.01 given the ACAOD perturbation of -20% and -10%, respectively. On the other hand, overestimated ACAOD yields lesser errors of +0.008, +0.015, and +0.03 in SSA following the ACAOD perturbation of +10%, +20%, and +40%, respectively. The SSA retrievals in the visible domain (470 nm) also responded to the perturbation in ACAOD, resulting in an error in SSA of -0.037, -0.017, and -0.007 given the ACAOD uncertainty of -40%, -20%, and -10%; the errors in SSA were +0.007, +0.013, and +0.025 when ACAOD was overestimated in the same proportion. The larger sensitivity of SSA to the ACAOD in the near-UV domain reflects the spectral dependence of extinction produced by fine-mode carbonaceous aerosols, resulting in relatively higher (lower) AODs at shorter (longer) wavelengths, thereby producing a larger absolute change in ACAOD given fixed perturbation. Since ACAOD for fine mode smoke particles are higher at shorter wavelengths, perturbing ACAODs by, say, 20% at 388 nm would produce a larger absolute difference in ACAOD than that caused by the same perturbation (20%) in ACAOD at 470 nm.

The ACAOD measurements from airborne HSRL-2 are accurate within the absolute uncertainty of ~0.01-0.02, which is equivalent to that of the ground-based AERONET data. For the September 2016 operation of ORACLES, the HSRL-2 data of the R9 version reports the absolute uncertainty in ACAOD measurements at 355 and 532 nm, which for the former wavelength was in the range of 0.8%-1.8%. Such uncertainty estimates for the remaining two years of HSRL-

2 operation in 2017 and 2018 are not yet available. Assuming similar levels of uncertainty during all three years of HSRL-2 operation, an error of 2% in the measured ACAOD would produce an error of ±0.002 in the retrieved SSA at 388 nm and 470 nm. The 4STAR dataset of altitude-dependent AOD collected during all three years of the ORACLES campaign provides absolute uncertainty for each instance of AOD measurement. The overall uncertainty in the 4STAR AOD measurements (>0.2 at 500 nm) collocated with OMI and MODIS was in the range of 1%-18% with mean values of 4.3% and 4.8%, respectively.

An additional source of uncertainty in using the 4STAR AOD is the vertical adjustment of altitude-dependent AODs to the observed cloud top. While we use actual AOD values measured at the corresponding altitudes adjusted to the cloud top using an average AOD profile derived for a specific flight (i.e., Figure 3), any departure of the actual profile from the averaged profile would yield erroneous AODs at the cloud top. The magnitude of error in vertically adjusted AODs depends on the variability of aerosol vertical distribution in spatial areas covered by the aircraft. It is hard to quantify this part of the error in AOD due to the lack of vertically resolved aerosol measurements along the P3/4STAR tracks. Assuming an overall 10% error in the measured AOD, including absolute error in the measurements, imperfectness in spatial-temporal collocation, and non-representativeness averaged AOD profile, would result in an error of less than 0.01 in the retrieved SSA at both wavelengths.

### 7.2    ERROR IN SSA DUE TO UNCERTAIN AEROSOL LAYER HEIGHT
The proposed synergy algorithm also involves an assumption of the aerosol vertical profile and mean layer height above the cloud deck. In the SSA inversion, the detailed vertical distribution of extinction measured by HSRL-2 for each collocated measurement was used to calculate the extinction-weighted mean aerosol layer height and assigned to the corresponding OMI/MODIS retrieval instances. The 4STAR sensor does not provide detailed, full-column vertical measurements of aerosols. Therefore, in the absence of such information for 4STAR-satellite collocated measurements, the algorithm relied on the ALH database adopted in the OMI standard OMAERUV aerosol algorithm. The OMI ALH dataset derived from the 30-month long collocated OMI-CALIOP measurements in clear-sky conditions (Torres et al., 2013) is expected to be accurate within ±1 km. The corresponding errors in the present above-cloud SSA retrieval obtained following the perturbation approach are found to be relatively lower, where an underestimated (overestimated) ALH by 1 km produces an error of -0.008 (-0.005) and -0.0017 (0.00016) at 388 nm and 470 nm wavelengths, respectively. Relatively larger errors in the retrieved SSA at near-UV wavelength signify the radiative interactions between the molecular Rayleigh scattering and aerosols-clouds, which gradually decrease at longer wavelengths, thereby producing smaller errors in the retrieved SSA at longer wavelengths.

### 7.3    ERROR IN SSA DUE TO UNCERTAIN ABSORPTION ÅNGSTRÖM EXPONENT (AAE)
Another important parameter that can affect the accuracy of the retrieved SSA is the value of Absorption Ångström Exponent (AAE) assumed in the aerosol model for both near-UV and VIS domains. AAE describes the spectral dependence of aerosol absorption and contributes to the 'color ratio' effect at TOA in addition to the magnitude of AOD and brightness of the cloud or COD. For instance, explaining two-channel TOA reflectance observations assuming two distinct values of AAE results in different AOD retrievals in clear-sky (Jethva and Torres, 2011) as well

as in the above-cloud aerosol situation (Jethva et al., 2018). The AAE for the near-UV aerosol model used in the present study was adopted from the standard OMI/OMAERUV carbonaceous aerosol models based on the above two studies. The smoke particles are assumed to be so-called 'colored aerosols', representing spectrally varying imaginary part of the refractive index, in which the latter is assumed to be 20% higher at 354 nm than that at 388 nm (see Table 2). This spectral dependence in the imaginary part of the refractive index translated into AAE in the range 2.45-2.60, which is expected to adequately represent the organic content of the carbonaceous smoke from biomass burning.

More recently, Kayetha et al. (2022) retrieved spectral SSA following the synergy between ground-based AOD and OMI-MODIS observations for the clear-sky scenes over more than 100 AERONET sites. The retrieved AAE values in the near-UV (354-388 nm) and visible domains (470-660 nm) over *Mongu* station in central/southern Africa representing biomass burning fine mode smoke particles were retrieved as 2.49 (3.02) and 0.92 (0.81), respectively, for the Jun-Jul-Aug (Sep-Oct-Nov) season. The corresponding spectral dependence in the retrieved imaginary part of the refractive index at the near-UV wavelengths was noted at ~ 13% and 20% for JJA and SON seasons. The AAE values (1.15-1.35) for the visible wavelength range (470-860 nm) are directly adopted from the ground-based AERONET inversion database.

For estimating errors in the retrieved SSA due to changes in the assumed AAE, the imaginary part of the refractive index at 354 nm and 860 nm was perturbed by ±10% while retaining the original values at 388 nm and 470 nm as described in the aerosol model (Table 2). The corresponding change in AAE in the near-UV and visible wavelength domains were noted ±0.7 and ±0.12, respectively. An increase (decrease) in the imaginary index at a shorter wavelength results in steeper (reduced) spectral dependence of aerosol absorption or AAE. The above-cloud aerosol LUTs were re-generated by performing RT calculations with a perturbed imaginary part of the refractive index. Other input parameters were unchanged and kept the same as described in the original aerosol model. The above-cloud SSA was retrieved using the perturbed aerosol LUT and compared against the original retrieval dataset derived from the OMI/MODIS-HSRL-2 synergy. For the near-UV domain, we find that changing the imaginary part of the refractive index at 354 nm by +10% (-10%) or AAE by +0.67 (-0.73) produces an error in the retrieved SSA by +0.028 (-0.02). In other words, an increment (decrement) in the assumed AAE value yields overestimation (underestimation) in the retrieved SSA above the clouds. Similar results were found for the visible domain (470-860 nm), albeit with much lower magnitudes of error in SSA (+0.0048, -0.005) resulting from the corresponding change in AAE. Larger errors in the retrieved SSA at 388 nm compared to those at 470 nm signify the greater sensitivity of TOA radiance to the aerosol absorption above the cloud in the near-UV region.

### 7.4    ERROR IN SSA DUE TO CHANGES IN PARTICLE SIZE DISTRIBUTION AND REAL PART OF REFRACTIVE INDEX

Finally, the errors in the retrieved SSA were quantified due to changes in the assumed aerosol particle size distribution. The aerosol model used in the present work was essentially developed using long-term AERONET direct and inversion measurements (see Table 2). Although we assume that the aerosol particle size distribution (PSD) model adequately represents the transported carbonaceous smoke above clouds over the Atlantic Ocean, the sensitivity of the retrieved spectral SSA to changes in the actual PSD from the assumed one, therefore, should be quantified. The

sensitivity of the retrieved spectral SSA is quantified by comparing the original retrieval dataset of 2016 OMI-MODIS-HSRL-2 synergy against the new dataset retrieved using the perturbed aerosol PSD models with +/- 20% and +/-5% perturbations in fine mode radius and real part of the refractive index, respectively.

An underestimation in the fine mode radius of 20%, i.e., from 0.0898 μm to 0.07184 μm (0.10776 μm), produces errors in SSA mostly lesser than +0.005 at 388 nm and 470 nm. On the other hand, an overestimated fine mode radius by 20% produces errors in SSA up to +0.03 with a mean value of +0.022 at 470 nm, whereas the error in SSA was noted lower than +0.01 at 388 nm. Similarly, an underestimation (overestimation) in the real part of the refractive index by 5% results in a mean error in SSA of +0.01 (-0.015) at 388 nm and +0.03 (-0.01) at 470 nm. Noticeably, a perturbation in the real part of the refractive index around its original value yields error in SSA in a symmetrical way, i.e., an underestimation and overestimation produce positive and negative errors, albeit the magnitude of errors is different, where overestimation in the real part of the refractive index results in larger positive errors.

## 7.5   ERROR IN SSA DUE TO NOT ACCOUNTING FOR NO$_2$

In the present inversion of above-cloud SSA, the trace gases were not accounted for explicitly, except that the columnar Ozone (O$_3$) amount was kept at a constant value of 275 Dobson Unit (DU) in the RT simulations for creating aerosol-cloud LUT. In addition to the spectral AODs above aircraft, 4STAR also measured the vertical column densities (VCD) of trace gases such as Ozone (O$_3$) and NO$_2$. The mean/median of vertical column densities (VCD) of O$_3$ measured by 4STAR onboard P3 for the September 2016 and August 2017 deployments were ~335 DU and ~267 DU, with a standard deviation of ~27 DU and ~13 DU, respectively. The assumed constant value of O$_3$ of 275 DU in the aerosol-cloud LUT is quite close to the mean amount measured by 4STAR in August 2017, but lower by 60 DU for September 2016. Since the wavelengths of the SSA retrieval from OMI (354 nm and 388 nm) and MODIS (470 nm and 860 nm) are known to be mostly unaffected by the O$_3$ absorption, we don't expect any significant error in the retrieved above-cloud SSA due to changes in the O$_3$ amounts.

To quantify the error in the retrieved SSA due to unaccounted NO$_2$, additional LUTs were generated assuming columnar NO$_2$ amounts of 0.25 DU and 0.50 DU embedded with aerosols and cloud parameters tabulated in Table 2. The retrievals of the above-cloud SSA were performed using these new LUTs and compared against the original OMI-HSRL-2 synergy inversion dataset to calculate the corresponding error in SSA caused by changing NO$_2$ amounts. We find that accounting for the presence of NO$_2$ by amounts 0.25 DU and 0.5 DU produces differences in the retrieved above-cloud SSA at OMI wavelength of 388 nm by -0.001 (±0.001) and -0.002 (±0.002), respectively. In other words, not accounting for the columnar NO$_2$ absorption effects, the retrieved SSA at 388 nm are subjected to an overestimation by the stated quantities given respective change in the columnar NO$_2$ amounts. For the blue band (470 nm) of MODIS, the retrieved SSAs were higher by +0.006 and +0.011 when considering NO$_2$ amounts of 0.25 DU and 0.50 DU, respectively, in the inversion.

The columnar NO$_2$ amounts commonly retrieved from Aura-OMI and Sentinel5p-TROPOMI sensors over the biomass burning regions of continental central/southern Africa range from 2-10 (x10$^{15}$ molecules/cm$^2$) or 0.07 DU to 0.37 DU (https://so2.gsfc.nasa.gov/no2/pix/regionals/Africa/Africa.html). However, due to its shorter lifetime and along the

westward transport pathways of smoke, the $NO_2$ VCD amounts over the ocean reduced significantly to 2-5 (x$10^{15}$ molecules/cm$^2$) or 0.07 DU to 0.19 DU (personnel communication: Lok Lamsal—$NO_2$ remote sensing expert at NASA Goddard Space Flight Center). Given such lower columnar amounts of $NO_2$ (<0.2 DU) over the ocean, the expected errors in the retrieved SSA above clouds are up to ~0.005 or lower.

### 7.6 ERROR IN SSA DUE TO CHANGES IN CLOUD EFFECTIVE RADIUS

The C1 cloud model adopted in the present application assumes a cloud effective radius of 12.0 μm based on the MODIS standard cloud product data over the Southeastern Atlantic Ocean. Though the histogram of CRE for the ORACLES period peaks around 12 μm, the retrieval dataset was distributed over a wide range of values ranging from 6-26 μm, albeit with 70%-80% of the data points falling within 6-18 μm. To quantify the sensitivity of the retrieved SSA to CRE, therefore, we perturbed the assumed value by ±6 μm, i.e., 6 μm and 18 μm, and created the aerosol-cloud LUTs for the retrieval of above-cloud SSAs. The resultant change in the retrieved SSA at 388 nm and 470 nm are listed in Tables 3 and 4. It is found that an underestimation (overestimation) of CRE by -6 μm (+6 μm) produced an error of +0.005 (-0.001) in the retrieved SSA at 388 nm. On the other hand, the changes in CRE by the same magnitude result in relatively larger errors of +0.023 (-0.012) in the retrieved SSA at 470 nm and 860 nm. It was shown using the radiative transfer simulations that the sensitivity of the spherical albedo or reflectance function of the liquid water cloud to changes in CRE is significantly smaller at shorter wavelengths (<0.5 μm), which gradually increases at longer wavelengths in the near-IR and shortwave-IR parts of the spectrum (King et al., 1997). Larger errors in the retrieved SSA at VIS-NIR wavelength caused by varying CRE signify the increased sensitivity of the reflected radiation to the droplet size at longer wavelengths.

The uncertainty matrix for the retrieved SSA at 388 nm and 470 nm of OMI and MODIS sensors, respectively, caused by prescribed error in the above-described input factors is listed in Table 3.

### 7.7 SENSITIVITY OF RETRIEVED COD TO CHANGES IN AEROSOL AND CLOUD PARAMETERS

The proposed inversion algorithm, along with the aerosol SSA, also co-retrieves aerosol-corrected COD. It is worthwhile to examine the errors in the retrieved COD due to perturbations in the assumed aerosol and cloud parameters, as it is carried out and discussed for the retrieved SSA in the above sub-sections. Tables 5 and 6 tabulate the % error in the retrieved COD caused by varying different aerosol parameters and cloud effective radius. To summarize the findings here, more realistic uncertainty in the input AODs of -20% (+20%) results in errors in the COD retrieval of +5% (-4%) at 388 nm, whereas the error at 860 nm was noted much smaller with magnitude -0.6% (+0.3%) with ~2% standard deviation around the mean values. A relatively larger error in COD at the near-UV wavelength results from a larger change in AOD given the fixed perturbation (in %) in the input AOD at a shorter wavelength compared to that at 470 nm. The reason for the reversal of sign in the error between 388 nm and 860 nm is not fully understood. Perturbing the ALH by ±1 km led to an error in COD of -2% to +3% at 388 nm, whereas the error for the 470 nm wavelength was significantly smaller (<0.5%). This is because of gradually reduced radiative interactions of aerosol-cloud and Rayleigh scattering in the VIS-NIR spectral domain, making ALH a less important assumption in the inversion. Changing the spectral dependence of absorption, quantified as AAE, resulted in an error

of -12% (+8%) in the retrieved COD at 388 nm, given an overestimation (underestimation) in AAE of +0.67 (-0.73). The corresponding errors in COD at 860 nm were lower by -1.5% (+1.4%).

The retrieved CODs in the near-UV region show some sensitivity to the aerosol PSD, where a change of mode radius by -20% (+20%) produced an error of +8% (-3%) in COD at 388 nm. The errors in COD at 860 nm were lower and about +3% (-3%). Similarly, the error in COD was within ±4% and <1% at 388 nm and 860 nm, respectively, due to ±5% change in the real part of the refractive index. Please note that the standard deviation of the error matrix was higher than the mean error reported here. Uncertainties in the retrieved COD due to not accounting for $NO_2$ in the inversion were -2.5% and -5% at 388 nm and -0.5% and -1% at 860 nm, for columnar $NO_2$ amount of 0.25 DU and 0.50, respectively.

Lastly, the errors in the retrieved COD are quantified due to the change in the cloud effective radius by ±6 μm from its originally assumed value of 12 μm. It is learned that an underestimated CRE by -6 μm produces about -9% and -13% in the retrieved COD at 388 nm and 470 nm, respectively. On the other hand, an overestimated CRE by +6 μm results in +2% and 6% uncertainty in COD at these two wavelengths. Restating here again, larger errors in the retrieved aerosol-corrected COD at NIR wavelength caused by varying CRE imply the enhanced sensitivity of the reflected radiation to the changing droplet size at longer wavelengths.

The uncertainty matrix (%) for the retrieved COD at 388 nm and 470 nm of OMI and MODIS sensors, respectively, resulted from the perturbation of the above-described input factors is listed in Table 4.

## 8    FINAL REMARKS

The 'color ratio' method designed to quantify aerosol loading above clouds using the TOA reflectance measurements from OMI and MODIS in the near-UV and VIS-NIR domains, respectively, assumes an aerosol model involving intrinsic optical properties of scattering and absorption. The theoretical RT simulations of aerosols above clouds showed an unambiguous, distinct sensitivity to the three main radiatively important parameters of aerosol-cloud overlap scene, i.e., ACAOD, COD, and SSA. Earlier studies noticed a marked sensitivity of the retrieved ACAOD from UV and VIS passive sensors to the assumed imaginary part of the refractive index or SSA. A perturbation of ±0.03 in the assumed aerosol SSA can lead to uncertainty in the retrieved ACAOD in the range -20% to +50% at the UV and VIS wavelengths, the variation of which depends on the magnitudes of both ACAOD and COD.

The RT simulations demonstrate that the availability of ACAOD from an external source, when constrained in the color ratio algorithm, allows the retrieval of the imaginary part of the refractive index of the lofted aerosol layer and the optical depth of the cloud underneath simultaneously.  The availability of high-quality, direct ACAOD measurements from airborne sensors HSRL-2 lidar and 4STAR sunphotometer acquired during the ORACLES campaign for 2016, 2017, and 2018 conducted over the southeastern Atlantic Ocean offered an unprecedented opportunity to test the proposed algorithm.

The retrieved above-cloud aerosol SSA at the near-UV (388 nm) and VIS (470 nm) wavelengths showed relatively lower bias with large variability (~0.8-0.95) at lower ACAOD (<0.2), above which the retrievals were more stable,

rendering no meaningful relation ACAOD for moderate to larger aerosol loadings. Under the lower aerosol loading conditions, the uncertainties in the assumed aerosol-cloud properties appear to dominate over the above-cloud aerosol signal, resulting in larger variabilities in the retrieved SSA.

A comparison of the retrieved SSA above the cloud at the visible wavelengths with the in-situ measurements collected from the PSAP + TSI nephelometer combination reveals an overall positive bias of ~0.01-0.02 in the satellite retrievals. One reason, among several others related to the inherent uncertainties in the satellite-based retrievals, could be the fact that the in-situ measurements correspond to the local sampling of smoke particles from instrumentation onboard, whereas the satellite retrievals represent columnar, ambient absorption properties of aerosols. The sensitivity analysis showed that lowering the assumed AAE in the aerosol model by 0.1-0.2 in the VIS-NIR spectral range could retrieve lower SSA (470 nm) by ~0.005-0.01; however, the change is not sufficient to explain the difference between in situ and satellite inversions.

A similar comparison against the airborne 4STAR inversion of SSA also reveals the tendency of satellite retrievals to be biased higher for the September 2016 ORACLES data. The differences were more pronounced at the near-UV wavelength of 388 nm than at the 470 nm wavelength. Changing the near-UV aerosol model with reduced spectral dependence of absorption or relatively lower AAE brought the observed RMSD down from 0.05 to 0.03—an improvement of 0.02 in SSA.

One important and promising result we achieved in this analysis is the consistency in measuring the intra-seasonal trend in the retrieved SSA of smoke transport above clouds over the southeastern Atlantic Ocean. The satellite retrievals, ORACLES in situ measurements, and ground-based inversion from AERONET all show increased absorption (lower SSA) in August, relatively weaker absorption in September, and even lower absorption (higher SSA) in October. Given the inherent uncertainties in all of these measurements, their relative biases, and the imperfections in the methodology, a consistent seasonal trend in SSA captured by different sensors run on distinct inversion methods is encouraging, which agrees with the previous studies (e.g., Eck et al., 2013). The level of agreement and consistency shown by the airborne-satellite synergy not only increases our confidence in the proposed inversion method but also motivates us to expand its application over other regions and even to a global scale, as briefly discussed in the following sub-section.

### 8.1 APPLICATION OF THE PROPOSED SYNERGY METHOD TO EXISTING LONG-TERM A-TRAIN RECORD

The synergy of airborne and satellite sensors demonstrated in the paper opens a possibility of synergizing the measurements from NASA's A-train satellite sensors to characterize aerosol absorption above the clouds in the UV-VIS spectral domain. For instance, the De-polarization Ratio (DR) method applied to the CALIOP 532-nm lidar backscatter measurements retrieves ACAOD at 532 nm without involving any assumptions on aerosol properties (Hu et al., 2007). The DR method is physically based on the two-way transmission of lidar signal and operates on scenes with opaque liquid water clouds with an assumed value of the cloud lidar ratio. The DR-based ACAODs retrieved from CALIOP for a couple of case studies of smoke aerosols above clouds agreed well with other passive satellite-based retrievals over the Southeastern Atlantic Ocean (Jethva et al., 2016). The DR method avoids any assumptions

on the intrinsic aerosol properties, whereas other satellite-based retrieval techniques of ACAOD based on passive measurements are sensitive to the chosen aerosol model. Therefore, the DR method is expected to deliver a more accurate quantification of aerosol loading above the cloud. As a part of NASA's A-train satellite constellation, the CALIOP, OMI, and MODIS sensors fly in formation and make measurements within a few minutes of time difference. The proposed synergy algorithm thus should be further tested and adopted for the potential A-train synergy to derive regional and even global, combined retrieval datasets of ACAOD, SSA, and aerosol-corrected COD. The availability of the joint aerosol-cloud product on a global scale will provide unprecedented quantitative information about the optical properties of aerosols and clouds, which play a crucial role in determining the sign and magnitude of the radiative effects of aerosols above clouds. Furthermore, the anticipated satellite synergy retrievals can help constrain the climate models with the much-needed observational estimates of the radiative effects of aerosols in cloudy regions and expand our ability to study aerosol direct and semi-direct effects on clouds and radiation balance.

## 8.2 FUTURE IMPLICATIONS

The Atmosphere Observing System (AOS) mission (https://aos.gsfc.nasa.gov/mission.htm) planned and partnered by several international space agencies, including NASA, will collect measurements of aerosols, clouds, convection and precipitation from a variety of instruments from Earth orbiting satellites and suborbital platforms. The AOS-Sky polar orbiting satellite is expected to be launched in 2031 carrying five instruments, including a backscatter lidar with measurement capability of aerosols and clouds at 532 nm and 1064 nm. The De-polarization Ratio method briefly discussed above and currently applicable to the CALIPSO observations would be a potential application to the AOS lidar measurements for the retrievals of above-cloud AOD that can be synergized with the contemporary observations from imaging radiometer for derivation of above-cloud aerosol SSA.

The Earth Cloud Aerosol and Radiation Explorer or EarthCARE satellite mission (https://earth.esa.int/eogateway/missions/earthcare#) of the European Space Agency (ESA) will carry an atmospheric lidar (ATLID) at the wavelength of 355 nm with a high-spectral resolution receiver and depolarization channel. The ATLID high spectral resolution lidar will provide the vertical distribution of direct extinction of aerosols without assuming lidar ratio, similar to how NASA's HSRL-2 lidar measures aerosols. Such direct measurements, when combined with the observations from Multi Spectral Imager (MSI), one of the four instruments planned onboard EarthCARE, will make possible the derivation of the above-cloud aerosol SSA following the inversion method demonstrated in this paper. Of course, future suborbital airborne campaigns with targeted measurements of aerosols and clouds in the same atmospheric column, such as demonstrated with ORACLES airborne data in the present work, would provide additional opportunities to test and apply the synergistic approach for characterizing the aerosol absorption in cloudy atmosphere.

## AUTHOR CONTRIBUTION

Dr. Jethva conceptualize the work, carried out inversion procedure, data analysis, graphics preparation, and wrote the manuscript. Dr. Torres overall supervised the workflow. R. Ferrare, S. Burton, A. Cook, D. Harper, and C. Hostetler helped in interpretating and using the HSRL-2 dataset. J. Redemann, S. LeBlanc, and K. Pistone, L. Mitchell, C. Flynn landed support in using the 4STAR direct and inversion aerosol datasets implemented in the present study. V. Kayetha helped in the discussion and editing the manuscript. All authors have contributed to improving the writeup and the content of the manuscript.

## CODE AND DATA AVAILABILITY

The code and results shown in the present paper can be obtained from the first authors upon request. The ORALCES airborne measurements used in the study can be accessed directly from NASA's ESPO web portal https://espo.nasa.gov/ORACLES/content/ORACLES. The Aura-OMI OMACA product containing L1B data used in this study was obtained from the Aura Validation Data Center at https://avdc.gsfc.nasa.gov/. The Terra-Aqua/MODIS L1b datasets were acquired from the Level-1 and Atmosphere Archive & Distribution System (LAADS) Distributed Active Archive Center (DAAC), located in the Goddard Space Flight Center in Greenbelt, Maryland (https://ladsweb.nascom.nasa.gov/).

## COMPETING INTERESTS

At least one of the (co-)authors is a member of the editorial board of Atmospheric Measurement Techniques.

**ACKNOWLEDGMENTS**

The authors acknowledge the efforts of the ORACLES team, both deployment support staff and the science team, for
successfully conducting airborne measurements during the 2016 and 2017 campaigns and freely distributing the
valuable datasets to the user community on NASA's ESPO web portal https://espo.nasa.gov/oracles.
Acknowledgments are also due to the MODIS Adaptive Processing System (MODAPS) and OMI SIPS teams
(https://earthdata.nasa.gov/about/sips/sips-modaps) for processing MODIS and OMI L1B datasets used in the present
study. The Terra-Aqua/MODIS L1b datasets were acquired from the Level-1 and Atmosphere Archive & Distribution
System (LAADS) Distributed Active Archive Center (DAAC), located in the Goddard Space Flight Center in
Greenbelt, Maryland (https://ladsweb.nascom.nasa.gov/). The Aura-OMI OMACA product containing L1B data used
in this study was obtained from the Aura Validation Data Center at https://avdc.gsfc.nasa.gov/. The AERONET-OMI-
MODIS synergy SSA retrieval dataset for the Mongu site in southern/central Africa used in the study for comparison
was obtained from Kayetha et al. (2022). We thank all PI(s) and Co-I(s) and their staff for establishing and maintaining
the AERONET sites over central/southern Africa whose datasets are used for the comparison in this study. The present
research work was funded by the NASA ROSES-2019 Aura Science Team Call (Solicitation: A.22 NNH19ZDA001N-
AURAST) under the grant number NASA Award/Grant Number: 80NSSC20K0946.

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

**Table 1** Airborne-satellite sensors and their respective datasets used in the present study.

| Platform/ Instrument | Operation Period | Measurement Characteristics |
|---|---|---|
| ER2/HSRL-2 | ORACLES Phase I (Aug-Sep 2016)[1] | Detailed vertical profile of aerosol backscatter and extinction Above-cloud column aerosol optical depth at 355 nm and 532 nm Dataset version: R9 (2016), R1 (2017), R1 (2018) |
| P3-Orion/4STAR P3-Orion/HSRL-2 | ORACLES Phase 1 (Aug-Sep 2016)[1] Phase 2 (Aug-Sep 2017)[2] Phase 3 (Sep-Oct 2018)[3] | Direct Sun measurements of spectral aerosol optical depth 4STAR dataset version: R4 (2016), R1 (2017), R1 (2018)  4STAR sky scan inversion of spectral SSA R1 (2016), R0 (2017) |
| P3-Orion/PSAP+TSI Nephelometer | Phase 1 (Aug-Sep 2016)[1] Phase 2 (Aug-Sep 2017)[2] Phase 3 (Sep-Oct 2018)[3] | HiGEAR onboard in situ measurements of aerosol SSA Version: R2 for all three years |
| Aura/OMI | Collocated in space and time with ORACLES observations | L1B top-of-atmosphere reflectance at 354 nm and 388 nm at nadir pixel resolution of 13x24 km²  OMACA Level-2 product for identification of above-cloud aerosols scene |
| Terra-Aqua /MODIS | Collocated in space and time with ORACLES campaigns | L1B top-of-atmosphere reflectance at 470 nm and 860 nm (MOD/MYD021KM) Geolocation and geometry (MOD/MYD03) L2 cloud product (MOD/MYD06_L2)  1-km measurements averaged to 5-km nominal grid |

[1]ORACLES-1 Aug-Sep 2016
4STAR on P3-Orion flights (Total 12): Aug 27, 30, 31; Sep 02, 08, 10, 14, 18, 20, 24, 25, 27
HSRL-2 on ER-2 flights (total 7): Aug 26, Sep 12, 16, 18, 20, 22, 24

[2]ORACLES-2 Aug-Sep 2017
HSRL-2/4STAR on P3-Orion flights (Total 15): Aug 09, 12, 13, 15, 17, 18, 19, 21, 24, 26, 28, 30, 31; Sep 02, 03

[3]ORACLES-3 Sep-Oct 2018
HSRL-2/4STAR on P3-Orion flights (Total 15): Sep 24, 27, 30; Oct 02, 03, 05, 07, 10, 12, 15, 17, 19, 21, 23, 25


**Table 2** Optical-microphysical properties of the carbonaceous aerosol model and radiative transfer configurations adopted in the synergy inversion algorithm. The model numbers are ordered from most absorbing to most scattering aerosols. AAE represents the Absorption Ångström Exponent for the UV (354-388 nm) and VIS-NIR (470-860 nm) wavelength domains.

| Model Number | Imaginary Index | | | | Single-scattering Albedo | | | | AAE | AAE |
|---|---|---|---|---|---|---|---|---|---|---|
| | 354 nm | 388 nm | 470 nm | 860 nm | 354 nm | 388 nm | 470 nm | 860 nm | 354-388 nm | 470-860 nm |
| 1 | 0.054 | 0.045 | 0.050 | 0.042 | 0.7729 | 0.7965 | 0.7635 | 0.6769 | 2.45075 | 1.2799 |
| 2 | 0.048 | 0.040 | 0.045 | 0.037 | 0.7914 | 0.8135 | 0.7805 | 0.7003 | 2.47316 | 1.2809 |
| 3 | 0.042 | 0.035 | 0.040 | 0.032 | 0.8109 | 0.8312 | 0.7983 | 0.7257 | 2.49355 | 1.2877 |
| 4 | 0.036 | 0.030 | 0.035 | 0.026 | 0.8316 | 0.8499 | 0.8171 | 0.7597 | 2.51118 | 1.3447 |
| 5 | 0.030 | 0.025 | 0.030 | 0.022 | 0.8535 | 0.8696 | 0.8369 | 0.7848 | 2.52527 | 1.3380 |
| 6 | 0.024 | 0.020 | 0.025 | 0.018 | 0.8769 | 0.8905 | 0.8578 | 0.8125 | 2.53511 | 1.3380 |
| 7 | 0.018 | 0.015 | 0.020 | 0.015 | 0.9020 | 0.9129 | 0.8802 | 0.8353 | 2.54113 | 1.2692 |
| 8 | 0.012 | 0.010 | 0.015 | 0.012 | 0.9293 | 0.9373 | 0.9043 | 0.8603 | 2.55022 | 1.1695 |
| 9 | 0.006 | 0.005 | 0.010 | 0.008 | 0.9602 | 0.9648 | 0.9308 | 0.8980 | 2.59644 | 1.1536 |
| 10 | 0.000 | 0.000 | 0.005 | 0.004 | 1.0000 | 1.0000 | 0.9612 | 0.9429 | N/A | 1.1566 |

Real refractive index = 1.5

Mean Radius (fine mode) = 0.0898 µm          Mean Radius (coarse mode) = 0.9444 µm
Standard Deviation (fine mode) = 1.4896 µm   Standard Deviation (coarse mode) = 1.9326 µm

Nodes of above-cloud aerosol LUTs
ACAOD (500 nm) = [0.0, 0.1, 0.25, 0.50, 0.75 1.00, 1.50, 2.00, 3.0, 5.0]
COD (wavelength-independent) = [0, 2, 5, 10, 20, 30, 40, 50]
Extinction Ångström Exponent (354-500) = 1.48
Extinction Ångström Exponent (470-860) = 1.80
Solar Zenith Angle (7) = [0°, 10°, 20°, 30°, 40°, 50°, 60°]
Viewing Zenith Angle (14) = [0°, 12°, 18°, 26°, 32°, 36°, 40°, 46°, 50°, 54°, 56°, 60°, 66°, 72°]
Relative Azimuth Angle (14) = [0°, 30°, 60°, 90°, 120°, 150°, 160°,165°,170°,175°,180°]
Aerosol Layer Height (3) = [3.0, 4.0, 5.0] in km
Surface Pressure Level = 1013.25 hPa
Surface Albedo (354-388) = 0.05
Surface Albedo (470-860) = 0.03


**Table 3** Mean uncertainties in the retrieved SSA (ΔSSA) at the 388 nm (OMI) and 470 nm (MODIS) due to errors in the algorithmic input parameters. Uncertainties are estimated by comparing the SSA retrievals from September 2016 OMI-HSRL2 collocated measurements derived assuming the perturbed input parameters against those retrieved from original, unperturbed input parameters.

| | ΔSSA due to change in AOD (388 nm/440 nm) ΔAOD | | | | | | | |
|---|---|---|---|---|---|---|---|---|
| | **-40%** | **-20%** | **-10%** | **-5%** | **+5%** | **+10%** | **+20%** | **+40%** |
| **388 nm** | -0.0536 | -0.0176 | -0.0081 | -0.0039 | 0.0039 | 0.0078 | 0.0154 | 0.0298 |
| **470 nm** | -0.0369 | -0.0156 | -0.0074 | -0.0036 | 0.0035 | 0.0069 | 0.0133 | 0.0247 |

| | ΔSSA due to change in ALH | | ΔSSA due to change in AAE ΔAAE corresponds to ±10% change in imaginary index at 354 nm and 860 nm | |
|---|---|---|---|---|
| | **ΔALH=-1 km** | **ΔALH=+1 km** | **ΔAAE=-0.73** | **ΔAAE=+0.67** |
| **388 nm** | -0.0077 | 0.0052 | -0.0198 | 0.0283 |
| **470 nm** | -0.0017 | 0.0002 | -0.0050 | 0.0048 |

| | ΔSSA due to change in Fine Mode Radius [μm] | | ΔSSA due to change in Real Part of Refractive Index | |
|---|---|---|---|---|
| | **-20%** | **+20%** | **-5%** | **+5%** |
| **388 nm** | -0.0010 | 0.004 | 0.012 | -0.008 |
| **470 nm** | 0.0209 | -0.004 | 0.022 | -0.018 |
| | ΔSSA due to not accounting for NO2 | | ΔSSA due to change in Cloud Effective Radius (μm) | |
| | **0.25 DU** | **0.50 DU** | **-6 μm (6 μm)** | **+6 μm (18 μm)** |
| **388 nm** | +0.001 | +0.002 | +0.005 | -0.001 |
| **470 nm** | -0.006 | -0.011 | +0.023 | -0.012 |

**Table 4** Mean uncertainties (%) in the retrieved aerosol-corrected COD (ΔCOD) at the 388 nm (OMI) and 470 nm (MODIS) due to errors in the algorithmic input parameters. Uncertainties are estimated in percentage by comparing the COD retrievals from September 2016 OMI-HSRL2 collocated measurements derived assuming the perturbed input parameters against those retrieved from original, unperturbed input parameters.

| | ΔCOD (388 nm) due to change in AOD (388 nm/470 nm) ΔAOD | | | | | | | |
|---|---|---|---|---|---|---|---|---|
| | **-40%** | **-20%** | **-10%** | **-5%** | **+5%** | **+10%** | **+20%** | **+40%** |
| **388 nm** | 12.80 | 4.54 | 2.02 | 0.99 | -0.96 | -1.92 | -3.84 | -7.66 |
| **470 nm** | -0.923 | -0.569 | -0.355 | -0.246 | 0.022 | 0.095 | 0.327 | 0.797 |

| | ΔCOD due to change in ALH | | ΔCOD due to change in AAE ΔAAE corresponds to ±10% change in imaginary index at 354 nm and 860 nm | |
|---|---|---|---|---|
| | **ΔALH=-1 km** | **ΔALH=+1 km** | **ΔAAE=-0.73** | **ΔAAE=+0.67** |
| **388 nm** | 2.930 | -2.18 | 7.63 | -11.95 |
| **470 nm** | 0.034 | -0.22 | 1.37 | -1.53 |

| | ΔCOD due to change in Fine Mode Radius [μm] | | Δ COD (388 nm) due to change in Real Part of Refractive Index | |
|---|---|---|---|---|
| | **-20%** | **+20%** | **-5%** | **+5%** |
| **388 nm** | 8.31 | -3.27 | 3.91 | -3.74 |
| **470 nm** | 2.72 | -2.46 | 0.84 | -0.78 |

| | ΔCOD (388 nm) due to not accounting for NO2 | | ΔCOD (388 nm) due change in Cloud Effective Radius (μm) | |
|---|---|---|---|---|
| | **0.25 DU** | **0.50 DU** | **-6 μm (6 μm)** | **+6 μm (18 μm)** |
| **388 nm** | -2.52 | -4.88 | -8.67 | +2.39 |
| **470 nm** | +0.50 | +0.99 | -13.29 | +6.17 |


**Figure 1.** Radiative transfer simulations of TOA reflectance in the a) near-UV (354-388 nm) and b) visible-near-infrared (470-860 nm) wavelength domains of an assumed atmosphere constituted by above-cloud carbonaceous aerosols with varying magnitudes of the imaginary part of the refractive index and COD. The simulation is carried out for the solar zenith, viewing zenith, and relative azimuth angles of 20°, 30°, and 120°, respectively, and assumed an aerosol layer with a quasi-Gaussian profile with peak concentration at 4 km above sea level placed above a half-kilometer thick cloud deck located between 850 hPa and 900 hPa. Three scenarios of aerosol loading are presented, i.e., ACAOD of 0.25 (left), 0.5 (center), and 1.0 (right) at 500 nm; ACAODs at other wavelengths were adjusted following the assumed spectral dependence of aerosol extinction. Solid (dotted) curves connecting simulations for the different values of the imaginary index (COD) represent sensitivity to aerosol absorption (cloud brightness).

**Figure 2.** Flight tracks of airborne HSRL-2 (a, c, e) and 4STAR (b, d, f) sensors onboard ER-2 and P3-Orion aircrafts color coded with the measured above-cloud AOD (500 nm) acquired during ORACLES-1 (top row), -2 (middle row), and -3 (bottom row) deployments in August-September 2016, August-September 2017, and September-October 2018, respectively. The histograms of the measured AOD are shown as insets in each plot. The dotted histograms of 4STAR (b, d, f) represent the measured AOD when the P3 aircraft flew just above the cloud top and beneath the aerosol layer above it, whereas the solid, black-colored histograms show all measurements.

**Figure 3.** Altitude-averaged vertical profiles (colored circles) of spectral AOD of the columnar atmosphere above the aircraft at 388 nm (red), 470 nm (blue), and 501 nm (green) wavelengths measured by 4STAR sunphotometer onboard P3-Orion flights for selected dates during the ORACLES-1 (top row), -2 (middle row), and -3 (bottom row) campaigns. AOD measurements were aggregated for each altitude grid interval of 0.1 km between 1.0 km and 4.0 km using the data for the entire flight to derive an averaged vertical profile and associated altitude versus AOD quadratic polynomial (solid lines). Coefficients of fitted quadratic relation for the three wavelengths are printed within each plot.

**Figure 4.** A simplified flowchart of the proposed airborne-satellite synergy algorithm for the retrieval of above-cloud aerosol SSA.

**Figure 5.** Spatial distribution of above-cloud aerosol SSA at 388 nm retrieved from the collocated measurements of OMI radiances and airborne HSRL-2 (left column) and 4STAR (right column) ACAOD (>0.2 at 500 nm) for the ORACLES-1 (top row-a, b), -2 (middle row-c, d), and -3 (bottom row-e, f) deployments. The corresponding histograms of the retrieved spectral SSA are also shown within each plot.

**Figure 6.** Same as in Figure 5 but for the above-cloud aerosol SSA at 470 nm retrieved from the collocated measurements of MODIS and HSRL-2/4STAR sensors.

**Figure 7.** Satellite-retrieved above-cloud aerosol SSA at 388 nm (top) and 470 nm (bottom) as a function of coincident airborne ACAODs used as input to the inversion algorithm. The data are represented as a box-whisker plot for each 0.1 bin size in ACAOD. The number of data points obtained for each bin of ACAOD is shown at the top/bottom of

the whisker. The satellite-airborne collocated datasets obtained from HSRL-2 and 4STAR for all three years of the ORACLES deployments are combined and included.

**Figure 8.** Spectral SSA retrieved from satellite-airborne synergy for the ORACLES-1 (top), -2 (middle), and -3 (bottom) campaigns. Retrievals are shown as a standard box-whisker plot format, where the mean and median are presented as filled circles and horizontal lines; shaded boxes represent data points contained between 25 and 75 percentiles; and thin vertical lines as 1.5 times interquartile range (75 minus 25 percentile). The black curves represent spline fit through two UV and three VIS-NIR wavelengths. Note that the 660-nm SSA retrievals represent interpolated values calculated using those at 470 nm and 860 nm. In-situ airborne measurements of SSA from PSAP and Nephelometer sensors for the respective ORACLES periods are shown as red circles, whereas spectral SSA inversions from AERONET over inland stations are depicted using different symbols in black. Retrievals of spectral SSA from a research technique combining ground-based AERONET AOD measurements and satellite observations from OMI and MODIS over the inland site *Mongu* in *Zambia* are shown as asterisks; the vertical dotted blue lines are the corresponding standard deviations.

**Figure 9** Comparison of spatiotemporally collocated above-cloud aerosol SSA retrieved from OMI (388 nm) against those derived from 4STAR sunphotometer sky scan observations for the ORACLES-1 September 2016 campaign. The OMI SSA retrievals using the original aerosol model (a, left) with 20% relative spectral dependence in the imaginary part of the refractive index (AAE ~2.45-2.60) and modified aerosol model (b, right) with 10% spectral dependence in the imaginary index (AAE ~1.72-1.87) are evaluated against those of 4STAR inversions. The OMI-4STAR matchups are color-coded according to the date of observations shown in legends. The arbitrary size of the circles represents the magnitude of coincident AOD (500 nm). The statistical measures of the comparison are printed in the lower left of each plot.

**Figure 10.** Comparison of spatiotemporally collocated above-cloud aerosol SSA retrieved from MODIS at 470 nm (left column) and 660 nm (right column) against those derived from 4STAR sunphotometer sky scan observations for the ORACLES-1 September 2016 (a, b) and ORACLES-2 August 2017 (c, d) campaigns. The MODIS-4STAR matchups are color-coded according to the date of observations shown in legends. The arbitrary size of the circles represents the magnitude of coincident AOD (500 nm). The statistical measures of the comparison are printed in the lower left of each plot.

**Figure 11.** A Comparison of the aerosol-corrected (y-axis) and non-corrected (apparent) cloud optical depths retrieved from the present synergy algorithm at 388 nm (red) and 860 nm (blue). The combined retrieval dataset derived from HSRL-2 and 4STAR synergy with OMI and MODIS for all three years of ORACLES campaigns (2016, 2017, and 2018) is used. The solid lines represent linear regression fits calculated from OMI (red) and MODIS (blue) retrievals. A similar comparison graph shown within the plot relates the aerosol-corrected cloud optical depth retrieved from the present work (y-axis) with that obtained from the OMI-OMACA standard product. b) % difference in the COD (retrieved-apparent) at 388 nm of OMI as a function of above-cloud aerosol absorption AOD (388 nm), where each

box represents 25 to 75 percentile data in AAOD bin size of 0.01. Different colors are used to denote data corresponding to different ranges of CODs. c) same as in (b) but for the 860 nm wavelength of MODIS.

**Figure 12.** Uncertainty estimates of derived SSA at 388 nm (left) and 470 nm (right) calculated by contrasting the original OMI-MODIS-HSRL-2 collocated retrievals for the 2016 campaign against those retrieved using perturbed ACAOD (red), assumed aerosol layer height (ALH, blue), and relative spectral dependence in absorption or AAE (green). The absolute change in SSA was calculated as perturbed minus original retrievals.

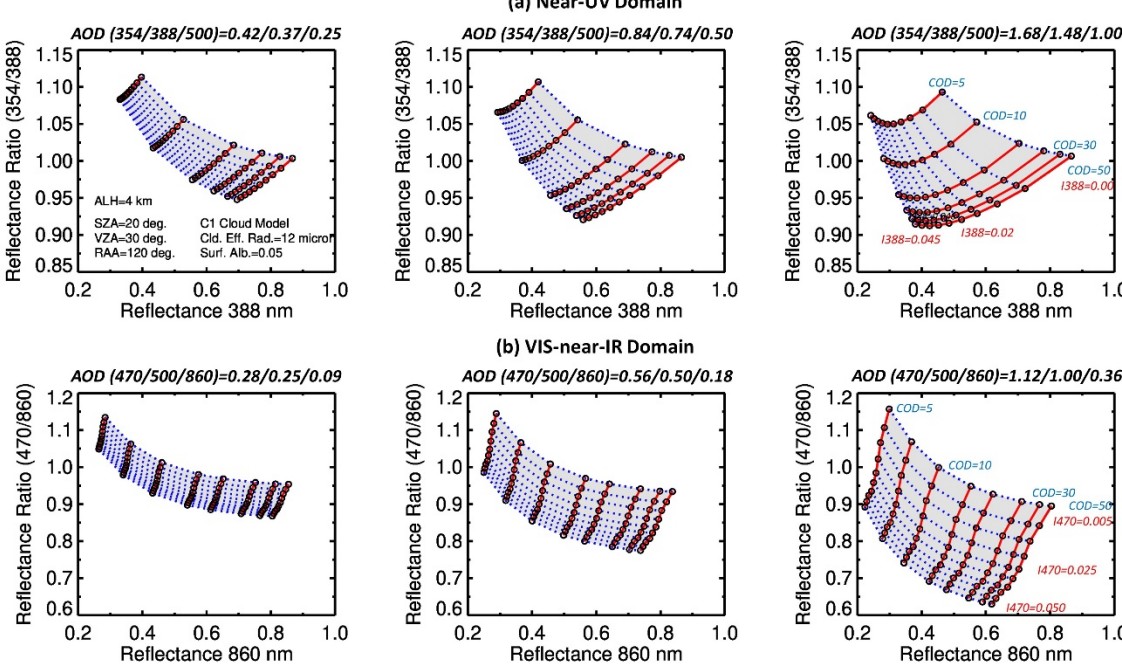

**Figure 1** Radiative transfer simulations of TOA reflectance in the a) near-UV (354-388 nm) and b) visible-near-infrared (470-860 nm) wavelength domains of an assumed atmosphere constituted by above-cloud carbonaceous aerosols with varying magnitudes of the imaginary part of the refractive index and COD. The simulation is carried out for the solar zenith, viewing zenith, and relative azimuth angles of 20°, 30°, and 120°, respectively, and assumed an aerosol layer with a quasi-Gaussian profile with peak concentration at 4 km above sea level placed above a half-

kilometer thick cloud deck located between 850 hPa and 900 hPa. Three scenarios of aerosol loading are presented, i.e., ACAOD of 0.25 (left), 0.5 (center), and 1.0 (right) at 500 nm; ACAODs at other wavelengths were adjusted following the assumed spectral dependence of aerosol extinction. Solid (dotted) curves connecting simulations for the different values of the imaginary index (COD) represent sensitivity to aerosol absorption (cloud brightness).

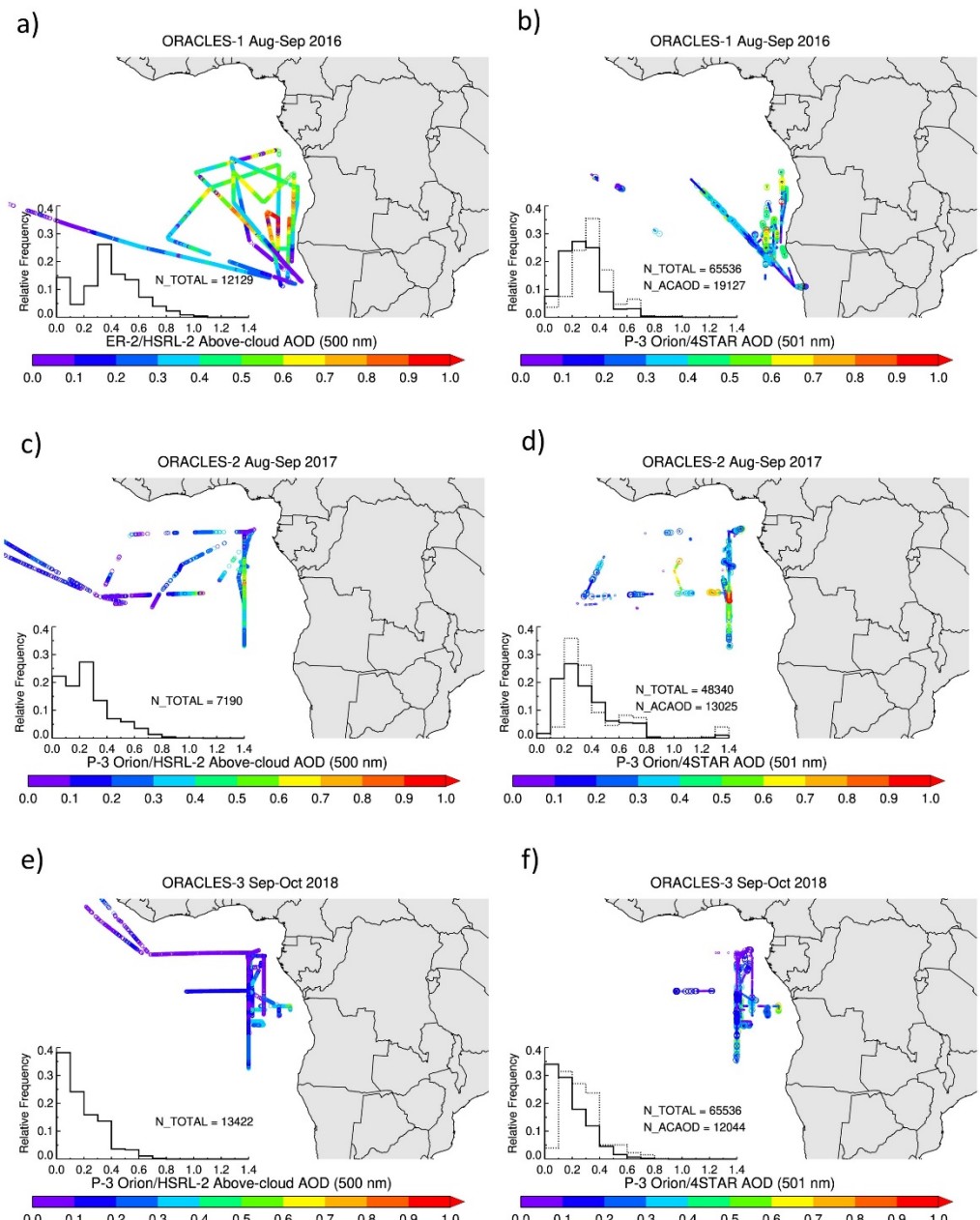


**Figure 2.** Flight tracks of airborne HSRL-2 (a, c, e) and 4STAR (b, d, f) sensors onboard ER-2 and P3-Orion aircrafts color coded with the measured above-cloud AOD (500 nm) acquired during ORACLES-1 (top row), -2 (middle row), and -3 (bottom row) deployments in August-September 2016, August-September 2017, and September-October 2018, respectively. The histograms of the measured AOD are shown as insets in each plot. The dotted histograms of 4STAR

(b, d, f) represent the measured AOD when the P3 aircraft flew just above the cloud top and beneath the aerosol layer above it, whereas the solid, black-colored histograms show all measurements.

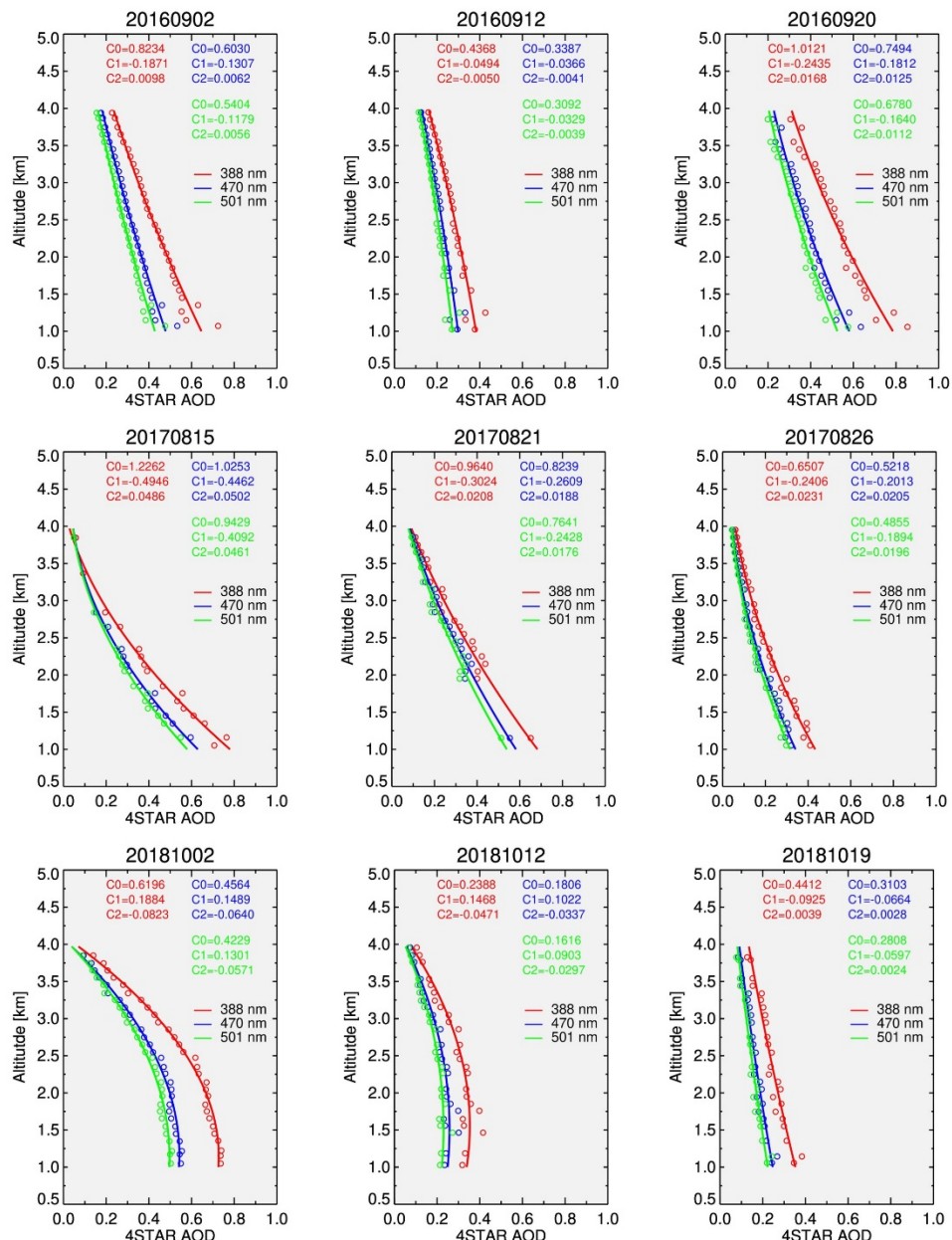

**Figure 3.** Altitude-averaged vertical profiles (colored circles) of spectral AOD of the columnar atmosphere above the
aircraft at 388 nm (red), 470 nm (blue), and 501 nm (green) wavelengths measured by 4STAR sunphotometer onboard
P3-Orion flights for selected dates during the ORACLES-1 (top row), -2 (middle row), and -3 (bottom row) campaigns.
AOD measurements were aggregated for each altitude grid interval of 0.1 km between 1.0 km and 4.0 km using the
data for the entire flight to derive an averaged vertical profile and associated altitude versus AOD quadratic polynomial
(solid lines). Coefficients of fitted quadratic relation for the three wavelengths are printed within each plot.

# Airborne-Satellite Above-Cloud Aerosol SSA Algorithm

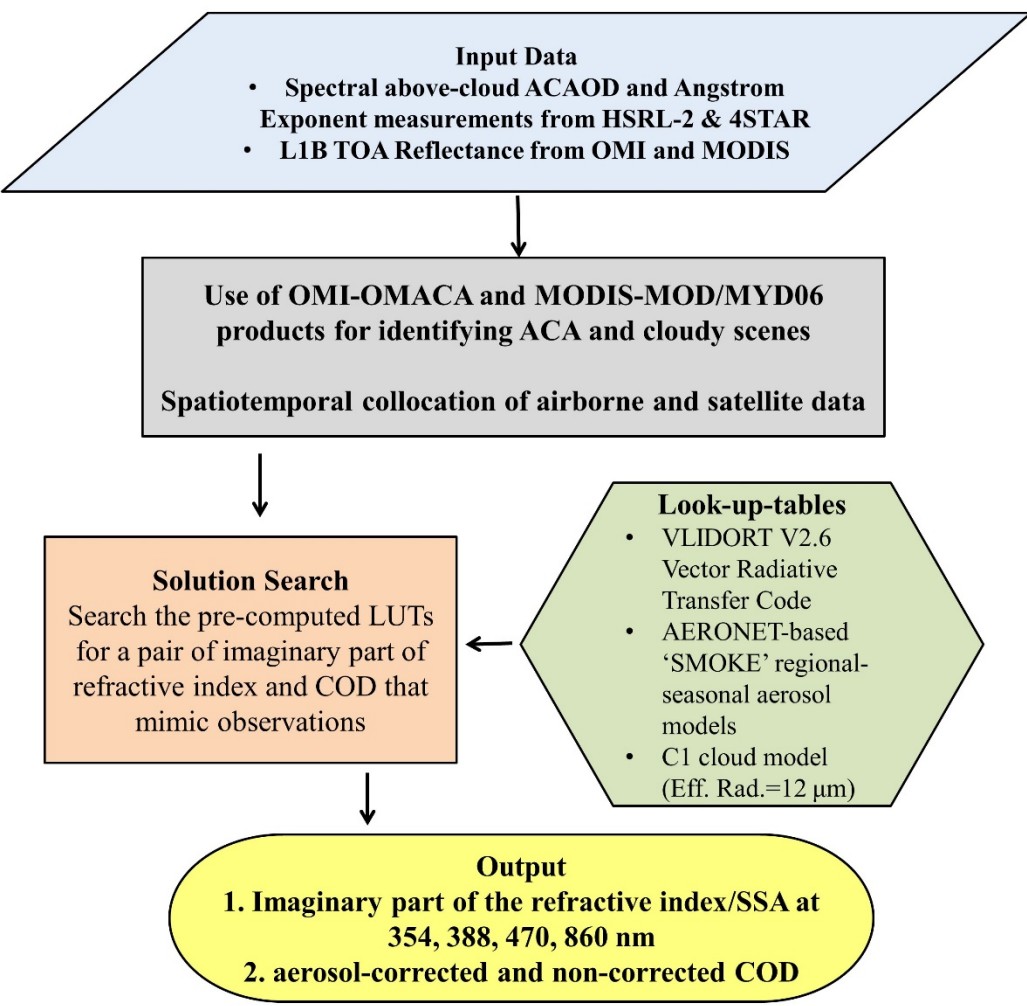


**Figure 4** A simplified flowchart of the proposed airborne-satellite synergy algorithm for the retrieval of above-cloud aerosol SSA.

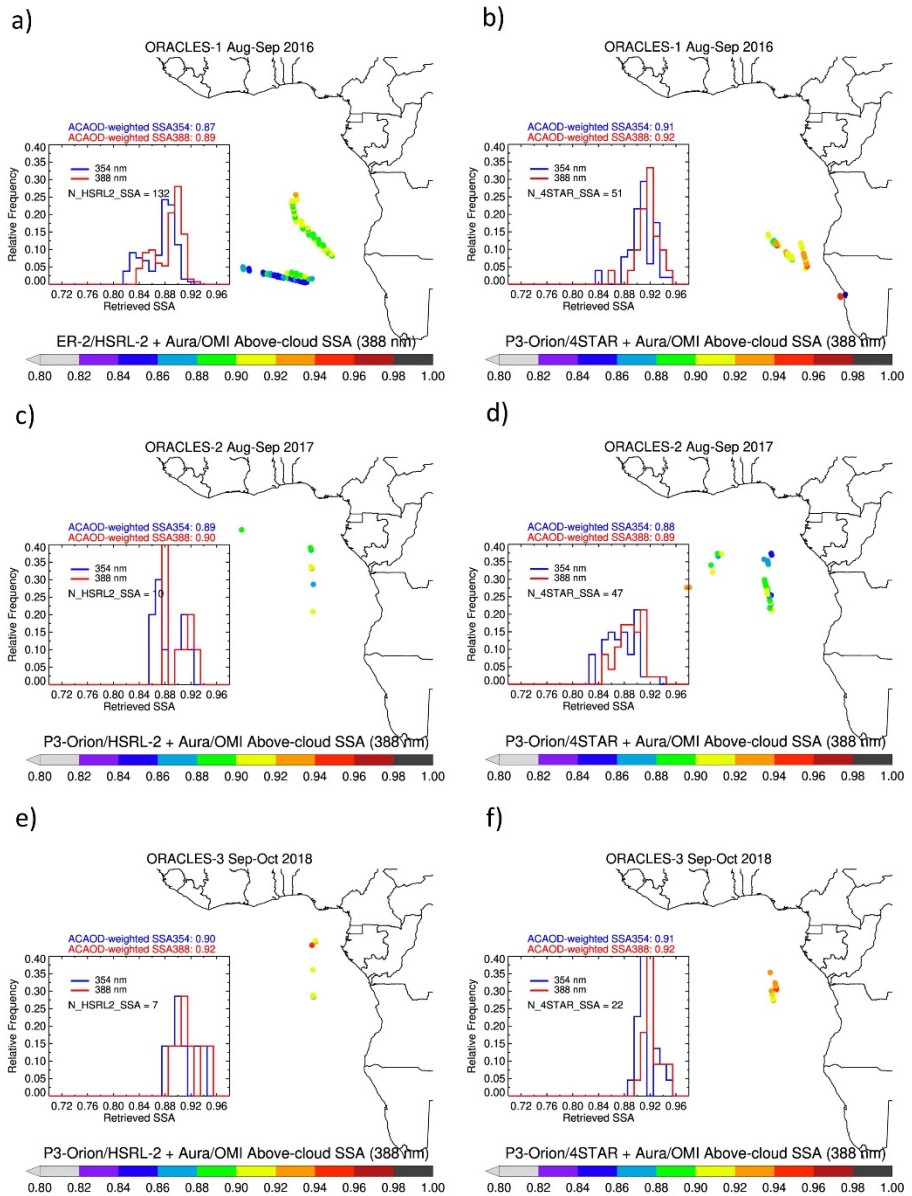

**Figure 5.** Spatial distribution of above-cloud aerosol SSA at 388 nm retrieved from the collocated measurements of OMI radiances and airborne HSRL-2 (left column) and 4STAR (right column) ACAOD (>0.2 at 500 nm) for the ORACLES-1 (top row-a, b), -2 (middle row-c, d), and -3 (bottom row-e, f) deployments. The corresponding histograms of the retrieved spectral SSA are also shown within each plot.

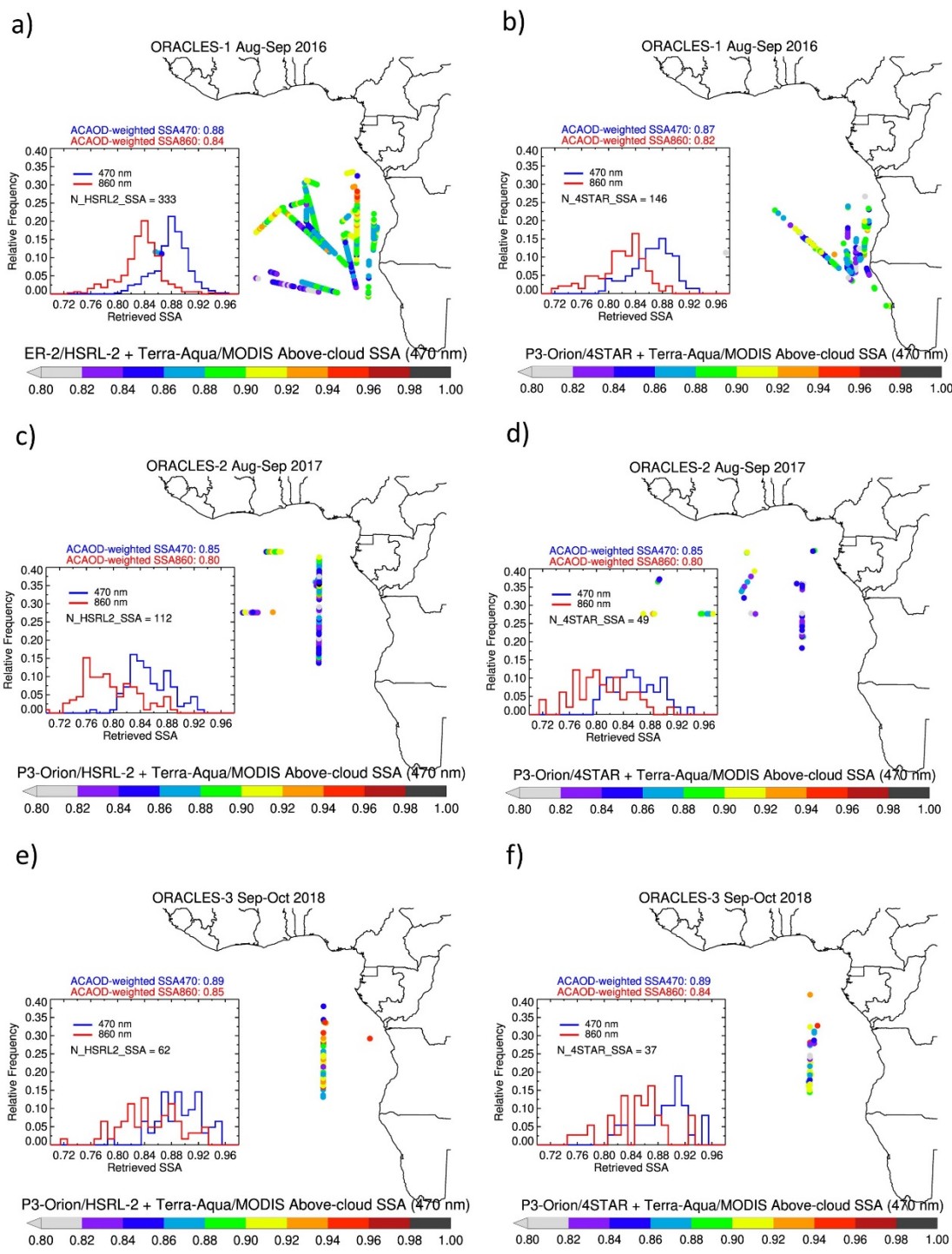


**Figure 6** Same as in Figure 5 but for the above-cloud aerosol SSA at 470 nm retrieved from the collocated measurements of MODIS and HSRL-2/4STAR sensors.

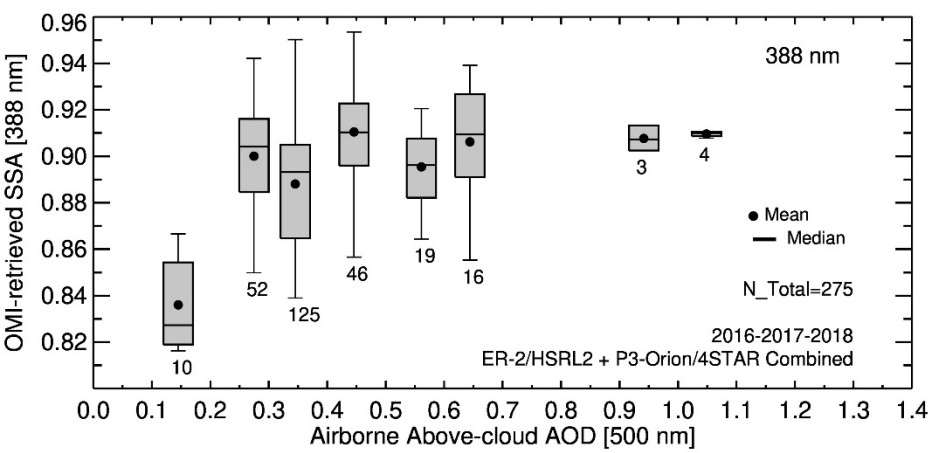

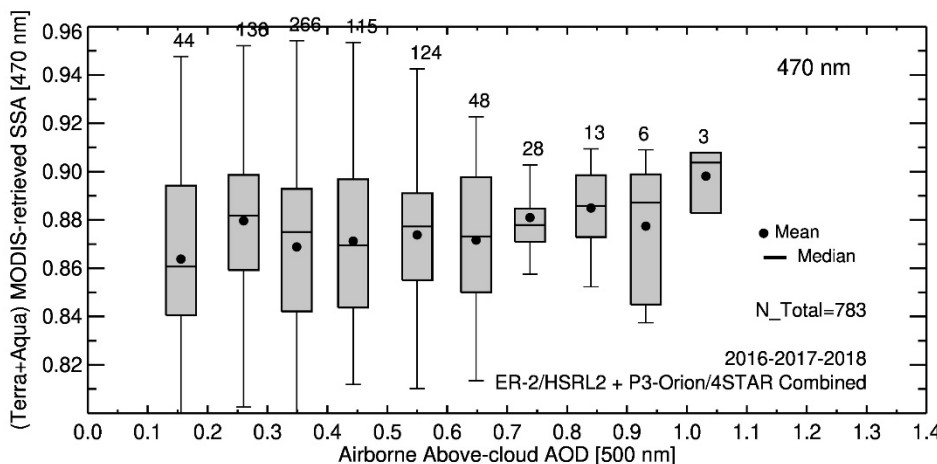

**Figure 7** Satellite-retrieved above-cloud aerosol SSA at 388 nm (top) and 470 nm (bottom) as a function of coincident airborne ACAODs used as input to the inversion algorithm. The data are represented as a box-whisker plot for each 0.1 bin size in ACAOD. The number of data points obtained for each bin of ACAOD is shown at the top/bottom of the whisker. The satellite-airborne collocated datasets obtained from HSRL-2 and 4STAR for all three years of the ORACLES deployments are combined and included.


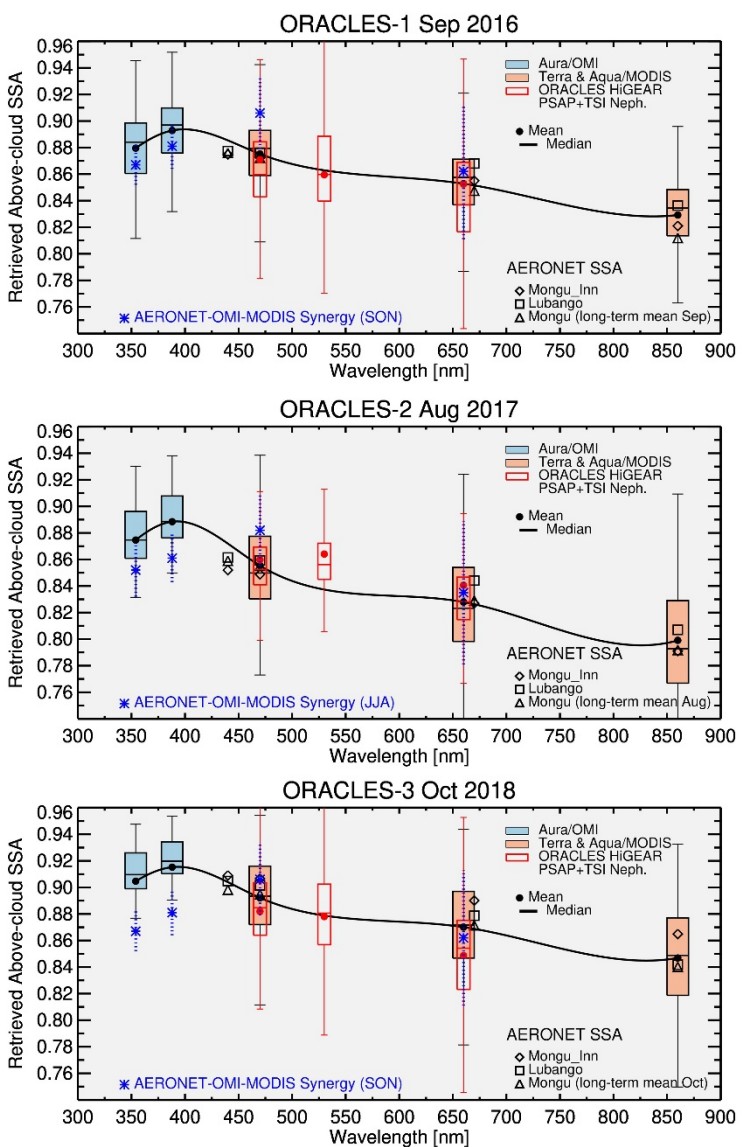

**Figure 8.** Spectral SSA above clouds retrieved from the combined OMI-MODIS synergy with both HSRL-2 and 4STAR for the ORACLES-1 (top), -2 (middle), and -3 (bottom) deployments. Retrievals are shown as a box-whisker plot format, where the mean and median are presented as filled circles and horizontal lines; shaded boxes represent data points contained between 25 and 75 percentiles; and thin vertical lines as 1.5 times interquartile range (75 minus 25 percentile). The black curves represent spline fit through two UV and three VIS-NIR wavelengths. Note that the 660-nm SSA retrievals represent interpolated values calculated using those at 470 nm and 860 nm. In-situ airborne measurements of SSA from PSAP and Nephelometer sensors for the respective ORACLES deployment years are shown as red box and whisker plots, and spectral SSA inversions from AERONET over inland stations are depicted using different symbols in black. Retrievals of spectral SSA from a research technique combining ground-based AERONET AOD measurements and satellite observations from OMI (354 nm and 388 nm) and MODIS (470 nm and 660 nm) over the inland site *Mongu* in Zambia are shown as blue asterisks; the vertical dotted blue lines are the corresponding standard deviations.

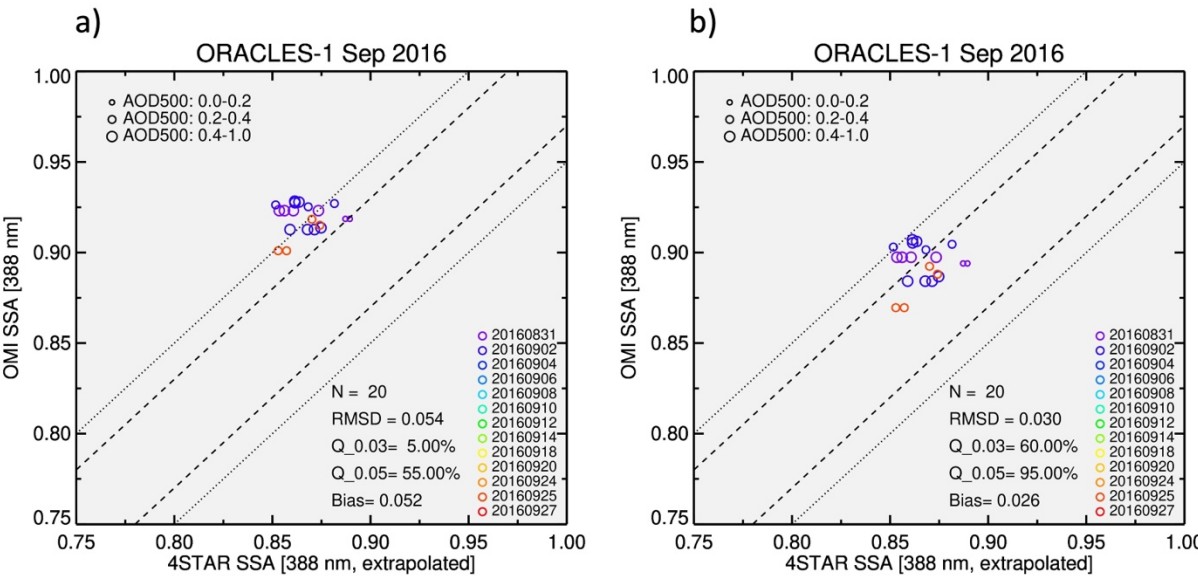

**Figure 9.** Comparison of spatiotemporally collocated above-cloud aerosol SSA retrieved from OMI (388 nm) against those derived from 4STAR sunphotometer sky scan observations for the ORACLES-1 September 2016 campaign. The OMI SSA retrievals using the original aerosol model (a, left) with 20% relative spectral dependence in the imaginary part of the refractive index (AAE ~2.45-2.60) and modified aerosol model (b, right) with 10% spectral dependence in the imaginary index (AAE ~1.72-1.87) are evaluated against those of 4STAR inversions. The Q_0.03

and Q_0.05 are the % matchups falling within the relative difference of ±0.03 and ±0.05, respectively. The OMI-4STAR matchups are color-coded according to the date of observations shown in legends. The sizing of the circles corresponds to the magnitude of coincident AOD (500 nm). The statistical measures of the comparison are printed in the lower left of each plot.


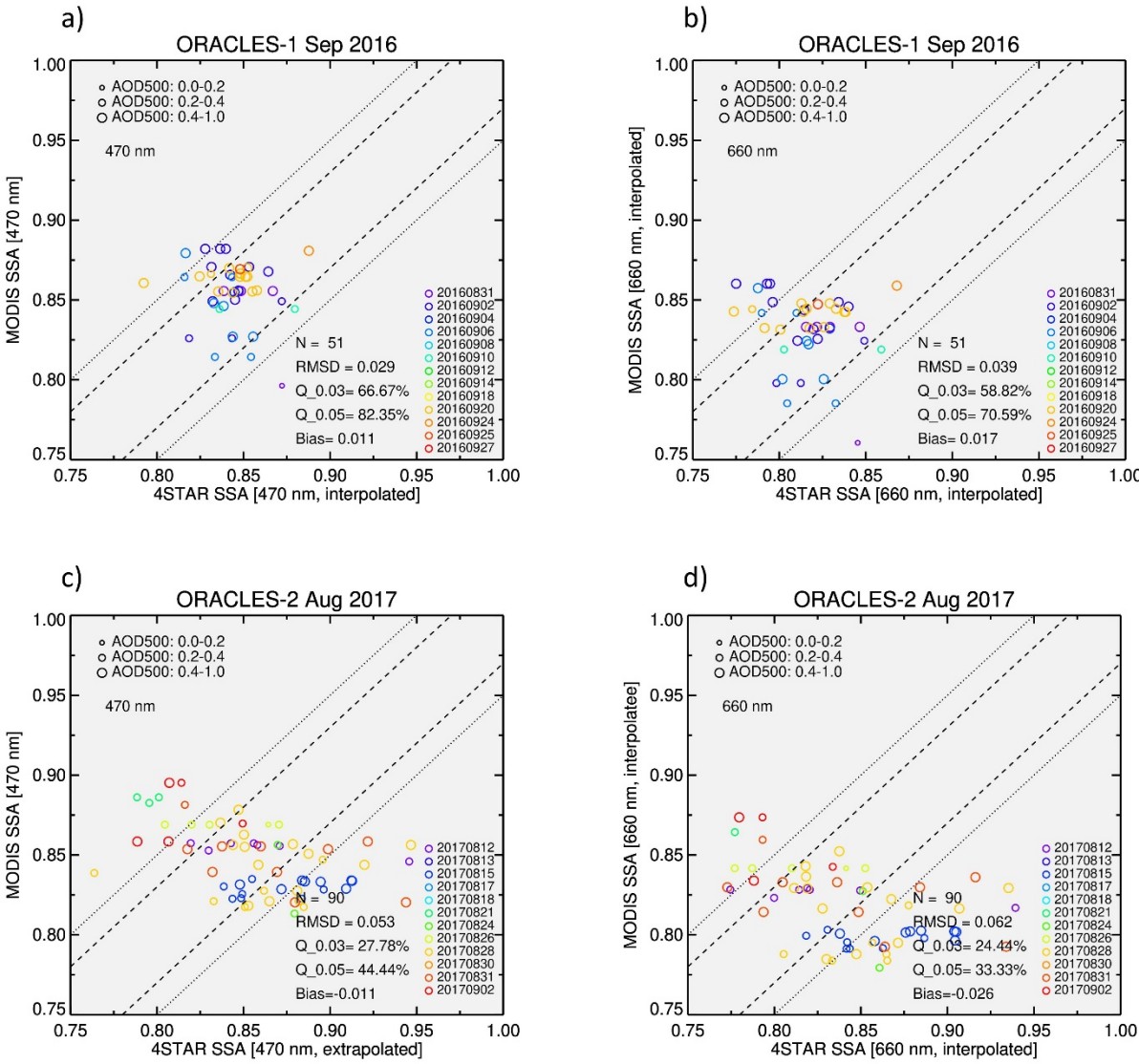

**Figure 10.** Comparison of spatiotemporally collocated above-cloud aerosol SSA retrieved from MODIS at 470 nm (left column) and 660 nm (right column) against those derived from 4STAR sunphotometer sky scan observations for the ORACLES-1 September 2016 (a, b) and ORACLES-2 August 2017 (c, d) campaigns. The Q_0.03 and Q_0.05 are the % matchups falling within the relative difference of ±0.03 and ±0.05, respectively. The MODIS-4STAR matchups are color-coded according to the date of observations shown in legends. The sizing of the circles corresponds to the magnitude of coincident AOD (500 nm). The statistical measures of the comparison are printed in the lower left of each plot.

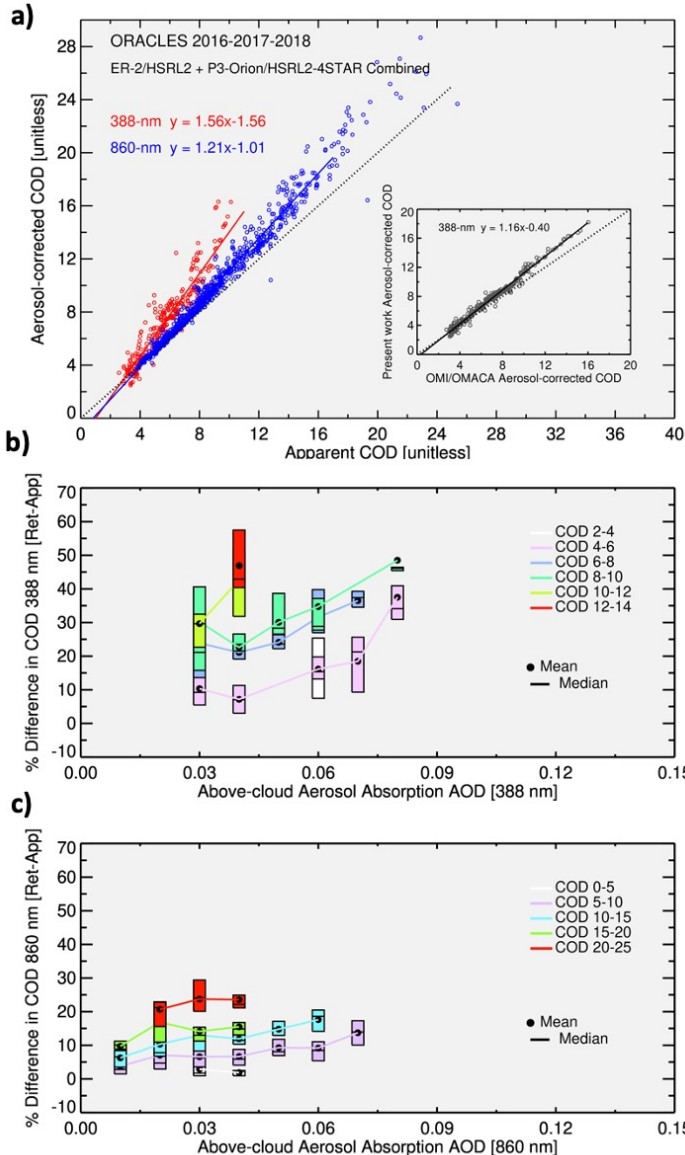

**Figure 11.** a) A comparison of the aerosol-corrected (y-axis) and non-corrected (apparent) cloud optical depths retrieved from the present synergy algorithm at 388 nm (red) and 860 nm (blue). The combined retrieval dataset derived from HSRL-2 and 4STAR synergy with OMI and MODIS for all three years of ORACLES campaigns (2016, 2017, and 2018) is used. The solid lines represent linear regression fits calculated from OMI (red) and MODIS (blue) retrievals. A similar comparison graph shown within the plot relates the aerosol-corrected cloud optical depth retrieved from the present work (y-axis) with that obtained from the OMI-OMACA standard product. b) % difference in the COD (retrieved-apparent) at 388 nm of OMI as a function of above-cloud aerosol absorption AOD (388 nm), where each box represents 25 to 75 percentile data in AAOD bin size of 0.01. Different colors are used to denote data corresponding to different ranges of CODs. c) same as in (b) but for the 860 nm wavelength of MODIS.

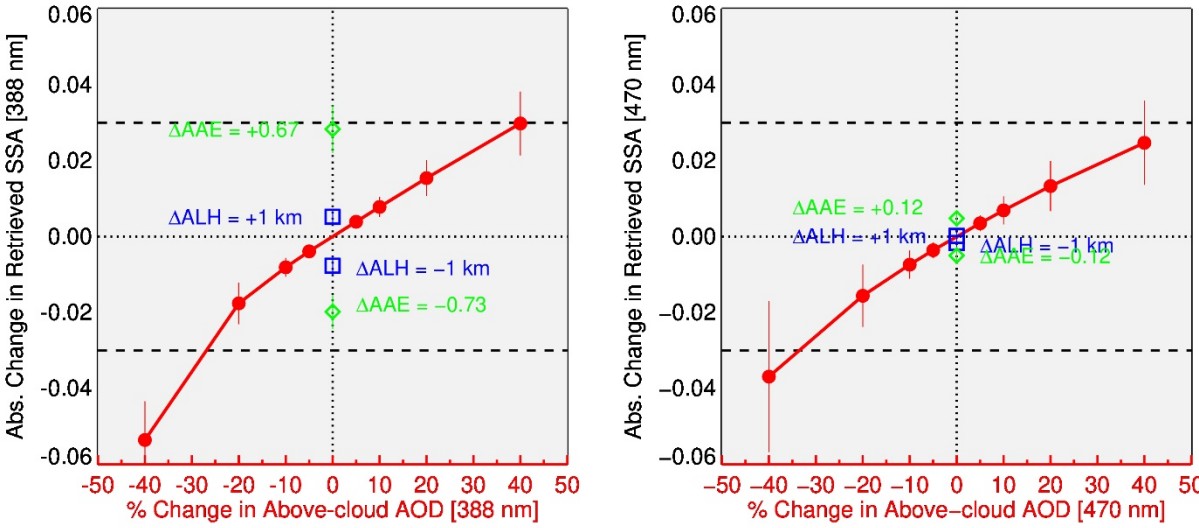

**Figure 12.** Uncertainty estimates of derived SSA at 388 nm (left) and 470 nm (right) calculated by contrasting the original OMI-MODIS-HSRL-2 collocated retrievals for the 2016 campaign against those retrieved using perturbed ACAOD (red), assumed aerosol layer height (ALH, blue), and relative spectral dependence in absorption or AAE (green). The absolute change in SSA was calculated as perturbed minus original retrievals.