# Peer review of "Retrieving UV-VIS Spectral Single-scattering Albedo of Absorbing Aerosols above Clouds from Synergy of ORACLES Airborne and A-train Sensors"

_EGUsphere, 2023_

## Referee Comment (RC1)

**General comments:** The authors developed a synergy algorithm that combines direct airborne measurements of above-cloud aerosol optical depth (ACAOD) and the top-of-atmosphere (TOA) spectral reflectance from NASA near-UV OMI and visible-NIR MODIS satellites to retrieve the single scattering albedo (SSA) of aerosols and the cloud optical depth (COD) for the scenarios in which light-absorbing aerosols are lofted above clouds. Applying the lidar (HSRL-2) and sunphotometer (4STAR) measured ACAOD during ORACLES field campaign in 2016-2018 over southeastern Atlantic Ocean and collocated OMI and MODIS observations into this algorithm, the SSA in both near-UV and visible-NIR for smoke above clouds were retrieved successfully with reasonable positive bias (0.01-0.02) by validating using airborne and in situ measurements. The differences of COD retrievals caused by smoke absorbing above clouds were also analyzed. Furthermore, the sensitivity analysis was done to quantify theoretical uncertainties in the retrieved SSA from errors in the measured ACAOD, aerosol layer height, and the ratio of the imaginary part of the refractive index (spectral dependence) of aerosols. I think this algorithm is new and first try to combine airborne and satellite data to retrieve SSA of aerosol above clouds, and could be extended to involve spaceborne lidar measurements to get global retrievals. It will be of interests to many readers of AMT. This manuscript is well organized and written, so I recommend it to be published after addressing the following specific comments.

**Specific comments**

1. Section 3.1: When the airborne measurements are introduced here, I suggest to add some descriptions about different flight modes of ER-2 and P3 since a few comparisons and different data processing methods are mentioned later. Actually, I don't fully understand the observation mode about how P3 aircraft profiled the atmosphere between the surface to 6 km. Did it only measure the atmosphere when flying vertically? What is the vertical resolution for the profiles?

2. Section 3.3.2: Should the "C1*Alt" in the first two equations here be the "C1*GPSAlt" and "C1*CldTopAlt", respectively?

3. Line 329-331: "The spectral dependence of the imaginary part of the refractive index in the visible-near-IR wavelength range (470-860 nm) is described by the AERONET dataset" is mentioned here, but it is unclear that how to describe this spectral dependence. Is any linear or exponential assumption used?

4. Line 342-344: It is mentioned here that "HSRL-2 was mounted on P3-Orion aircraft and measured the aerosol profiles below the aircraft level and above the cloud top". Since the aerosol vertical profile is assumed to follow a quasi-Gaussian distribution around the mean extinction-weighted aerosol layer height (ALH) from HSRL-2 measurements, if the peak height of this quasi-Gaussian distribution in reality is higher than the aircraft level where HSRL-2 cannot detect, ALH calculated from HSRL-2 measurements

will have larger difference with the peak height. Is this situation considered in the measurements?

5. Section 4.2: Some fixed microphysical parameters of cloud particles are mention here, including the effective radius, cloud layer height and cloud layer thickness. How will these parameters affect the aerosol-corrected COD retrieval? Given the effective radius comes from an old paper and the resources of cloud layer height and cloud layer thickness are not mentioned here, can they represent the climatology of clouds in the southeastern Atlantic Ocean? Maybe some references could be added.

6. Figure 6: Comparing Panel a and b, the SSA at 860 nm shows larger differences than that at 470 nm, even larger than other measurements in the rest of panels in this figure. Do you think what could be the possible reasons for this difference between HSRL-2 and 4STAR retrievals?

7. Section 7: The possible uncertainties in SSA retrievals from errors in aerosol properties are well analyzed in this section, but how will these properties affect aerosol-corrected COD? Similarly, how is the sensitivity of aerosol-corrected COD with respect to the cloud properties in the cloud model?

---

## Author Comment (AC1)

**egusphere-2023-1717**

**Retrieving UV-VIS Spectral Single-scattering Albedo of Absorbing Aerosols above Clouds from Synergy of ORACLES Airborne and A-train Sensors**

**Jethva et al.**

**Response to the Anonymous Referee # 1**

We thank and appreciate the anonymous reviewer for offering constructive and valuable comments & suggestions on our manuscript.

Before responding to each of the comments provided by the reviewer, we want to let both reviewers and the Editor know about the major upgrade we applied to the scientific content presented in our paper. The need to apply a significant revision stems from the comments made by referee # 1 regarding the cloud model properties. While responding to the comments, it was realized that the cloud effective radius assumed in the C1 cloud model adopted in the present work may not be an optimal representation of the stratocumulus clouds observed in the Southeastern Atlantic Ocean during the ORACLES operation period.

The figure below displays the histogram of the cloud effective radius (CRE) data derived from the standard Aqua/MODIS MYD06 cloud product over the Southeastern Atlantic Ocean from August through October 2016 to 2018. The CRE histograms for the months of August to October reveal a distinct peak around 11-12 μm with a wide distribution ranging from 6 μm to 26 μm, albeit 70%-80% of the data points are found to be within 6-18 μm. Meyer et al. (2015) have shown that the presence of partially absorbing carbonaceous smoke aerosols over clouds in this region has only a marginal impact on the retrieved CRE.

[Figure]

*Figure 1 Histograms of the Aqua/MODIS cloud effective radius retrievals over the Southeastern Atlantic for the ORACLES period of August through October 2016-2018.*

The C1 cloud model adopted in our work assumes the liquid droplet size distribution that follows the modified Gamma distribution with α, β, and γ parameters of 6.0, 1.5, and 1.0, respectively. The CRE value, when calculated using these parameters, of this cloud model was 6 μm, which is an underestimation compared to the MODIS CRE histograms discussed above.

Based on the MODIS CRE histograms corresponding to the region and period of interest in our study, we have revised the cloud model with an optimal CRE value of 12 μm. This is achieved by changing the α parameter from 6.0 to 15.0 and keeping the β and γ parameters unaltered.

**The entire analysis presented in the initially submitted manuscript was reperformed with the revised cloud model that required recreating the aerosol-cloud LUTs for the above-cloud SSA retrieval and its sensitivity analysis. The following are the significant changes we noticed in the results.**

1. The derived above-cloud SSAs are now lower in magnitudes by up to 0.01 and 0.02 at the near-UV and VIS-NIR wavelengths, respectively.
2. Most importantly, the comparison of the satellite-retrieved SSA against ORACLES in situ measurements, 4STAR inversions, and ground-based AERONET inversions improved noticeably, where the overestimation seen in the satellite retrievals is now either eliminated or largely reduced.
3. The results of the sensitivity analysis tabulated in Tables 3 and 4 show marginal change without its interpretation discussed in the original version of the manuscript.
4. The revised paper now also includes the uncertainty estimates in the retrieved aerosol-corrected COD resulting from perturbing the assumed aerosol and cloud parameters. These results are tabulated in Tables 5 and 6.

**All relevant figures and tables are updated with the revised calculations and results.**

The following are the one-to-one responses to each comment offered by referee # 1.

Referee comment **RC in RED**
Author's response **AR in BLACK**

**RC**: 1. Section 3.1: When the airborne measurements are introduced here, I suggest to add some descriptions about different flight modes of ER-2 and P3 since a few comparisons and different data processing methods are mentioned later. Actually, I don't fully understand the observation mode about how P3 aircraft profiled the atmosphere between the surface to 6 km. Did it only measure the atmosphere when flying vertically? What is the vertical resolution for the profiles?

**AR:** The ER-2 aircraft during the 2016 operation flew at about 20 km altitude and measured the vertically resolved aerosols and cloud properties for the atmospheric column below the aircraft level. On the other hand, P3 aircraft flew from the surface to about 5-6 km altitude, profiling the atmosphere vertically and horizontally within these altitude ranges. Different instruments onboard P3, including the 4STAR sunphotometer, made measurements both in vertical and horizontal dimensions along the aircraft trajectories.

Pertaining to the columnar AOD measurements above the P3 altitude, 4STAR made measurements at a vertical resolution in the range of 1-3 meters during all three deployments with an average resolution of ~2 meters. There were also instances during the flights when P3 flew horizontally without changing altitude and measured spectral AODs.

**RC:** 2. Section 3.3.2: Should the "C1*Alt" in the first two equations here be the "C1*GPSAlt" and "C1*CldTopAlt", respectively?

**AR:** That's true. '*Alt*' should be '*GPSAlt*' and '*CldTopAlt*' in the respective two equations. Thanks for pointing this out. The corrections are as follows.

$$AOD_{GPSAlt} = C0 + C1 * GPSAlt + C2 * GPSAlt^2$$

$$AOD_{CldTopAlt} = C0 + C1 * CldTopAlt + C2 * CldTopAlt^2$$

**RC:** 3. Line 329-331: "The spectral dependence of the imaginary part of the refractive index in the visible-near-IR wavelength range (470-860 nm) is described by the AERONET dataset" is mentioned here, but it is unclear that how to describe this spectral dependence. Is any linear or exponential assumption used?

**AR:** The AERONET Level 2.0 inversion algorithm retrieves both real and imaginary parts of the refractive index at four discrete wavelengths of 470 nm, 670 nm, 860 nm, and 1020 nm. Using the long-term inversion dataset of AERONET at an inland site, *Mongu,* for the biomass burning season (i.e., July, August, and September)*,* we calculated the ratio of the imaginary part of the refractive index between 470 nm and 860 nm. We use this 470/860 nm ratio of the imaginary index to constrain the spectral dependence of absorption between 470-860 nm. Table 2 lists the imaginary index values at 470 nm and 860 nm, in which the values at 470 nm were selected in intervals and the same at 860 nm were calculated based on the derived ratio from the long-term AERONET climatology dataset.

**RC:** 5. Section 4.2: Some fixed microphysical parameters of cloud particles are mentioned here, including the effective radius, cloud layer height and cloud layer thickness. How will these parameters affect the aerosol-corrected COD retrieval? Given the effective radius comes from an old paper and the resources of cloud layer height and cloud layer thickness are not mentioned here, can they represent the climatology of clouds in the southeastern Atlantic Ocean? Maybe some references could be added.

**AR:** Please refer to our response at the beginning of this report on the significant modification applied to the revision. Briefly mentioning here, we've changed the cloud effective radius (CRE) from 6 μm to 12 μm of the adopted C1 cloud model in accordance with the MODIS CRE retrievals over the Southeastern Atlantic Ocean. We've also quantified and added the uncertainty estimates in the retrieved SSA and aerosol-corrected COD due to perturbation in CRE by ±6 μm. Please refer to Tables 3 to 6 in the revised paper.

Regarding the cloud layer height and thickness, we looked at the CALIOP/CALIOP Level -2 cloud product over the study region and derived the mean and median values of cloud top and cloud base altitudes for 2016 and 2017. The table below shows that the cloud top detected by CALIOP was between 1.2-1.4 km, whereas the cloud base was between 0.5-0.6 km above sea level. In the present study, we assumed the cloud top and bottom altitudes of 1.5 km and 1.0 km, respectively. While the assumed cloud top altitude is in close agreement with CALIOP, the cloud bottom is slightly overestimated by ~0.4 km. However, we don't expect any noticeable changes in the derived SSA and COD retrievals due to a slight departure of the cloud bottom altitude from the observations.

We have added this description in the revised paper in section 4.2.

*Table 1 Cloud top and base altitudes (in km) over the Southeastern Atlantic Ocean (Lat: 33S to 0, Lon: 20W to 10E) detected by CALIOP/CALIPSO during biomass burning months of 2016 and 2017.*

| Months | Cloud Top (mean/median) [km] | Cloud Base (mean/median) [km] |
|--------|------------------------------|-------------------------------|
| August | 1.36/1.31 | 0.58/0.53 |
| September | 1.35/1.31 | 0.64/0.62 |
| October | 1.26/1.22 | 0.49/0.47 |

**RC:** 6. Figure 6: Comparing Panel a and b, the SSA at 860 nm shows larger differences than that at 470 nm, even larger than other measurements in the rest of panels in this figure. Do you think what could be the possible reasons for this difference between HSRL-2 and 4STAR retrievals? Do you think what could be the possible reasons for this difference between HSRL-2 and 4STAR retrievals?

**AR:** Note that we have updated all figures with the revised results derived from the modified cloud model. In the revised Figure 6, the spread in the retrieved SSA at both 470 nm and 860 nm is noticed to be larger with the 4STAR synergy (panel b) than with the HSRL-2 synergy (panel a).

The proposed algorithm was applied identically to both HSRL-2 and 4STAR measurements, accompanied with synergized OMI and MODIS TOA reflectance observations. The only difference between the two sets of results is the kind of ACAOD datasets, i.e., lidar vs. sun photometer. The HSRL-2 directly measured the columnar AOD above the cloud and used as an input to the synergy algorithm. On the other hand, 4STAR onboard P3 measured columnar AOD above the aircraft level, which required an altitude-based adjustment to the observed cloud-top (see section 3.3.2) in order to establish an equivalence with satellite measurements. The adjustment procedure used flight-averaged altitude vs. AOD polynomial to achieve this job. Any departure of the actual altitude vs. AOD profile for individual 4STAR-satellite matchups from the averaged representation, therefore, leads to erroneous columnar AOD above clouds and retrieved SSA. Another reason for a relatively larger spread in the 4STAR synergy retrievals of SSA could be the different aircraft trajectories, and therefore airmass, sampled by both airborne instruments that could result in different ranges of aerosol absorption properties in the region.

**RC:** 7. Section 7: The possible uncertainties in SSA retrievals from errors in aerosol properties are well analyzed in this section, but how will these properties affect aerosol-corrected COD?

**AR:** Thanks for pointing this out; we missed including the error matrix for COD due to changes in aerosol properties. In the revised manuscript, we have added additional Tables 5 and 6 containing the % error in the retrieved aerosol-corrected cloud optical depth at 388 nm and 470 nm, respectively, due to the presumed errors in input AOD, aerosol layer height, aerosol absorption Angstrom Exponent (AAE), particle size distribution, $NO_2$, and cloud effective radius.

Similarly, how is the sensitivity of aerosol-corrected COD with respect to the cloud properties in the cloud model?

To address the reviewer's concern, we also performed additional RT calculations of aerosols above cloud LUTs (both UV and VIS-NIR) with the C1 cloud model, but by perturbing the original cloud effective radius (CRE) of 12 μm by ±6 μm, i.e., 6 μm and 18 μm. The new set of aerosol-cloud LUTs was used to re-retrieve SSA and aerosol-corrected CODs for comparison with the original set of retrievals.

The changes in the retrieved SSA and COD are appended in Tables 3 through 6 in the revision. To summarize the findings, a decrease (an increase) in CRE by 6 μm produces a -8.67% (+2.39%) error in the retrieved COD at 388 nm. The errors in the retrieved COD at 860 nm, however, were higher and -13.29%

(6.17%). Larger (relatively smaller) errors in COD at longer (shorter) wavelength (388 nm and 860 nm in this case) imply greater (relatively weaker) sensitivity of the TOA reflectance and retrieved COD to the change in liquid water droplet distribution.

We've added a new sub-section 7.7 describing the sensitivity of the retrieved COD to various assumptions made in the inversion algorithm.

The new results are described in section 7 in the revision.

---

## Author Comment (AC2)

egusphere-2023-1717

**Retrieving UV-VIS Spectral Single-scattering Albedo of Absorbing Aerosols above Clouds from Synergy of ORACLES Airborne and A-train Sensors**

**Jethva et al.**

**Response to the Anonymous Referee # 2**

We thank and appreciate the anonymous reviewer for providing constructive and valuable comments on our manuscript.

Before responding to each of the comments provided by the reviewer, we want to let both reviewers and the Editor know about the major upgrade we applied to the scientific content presented in our paper. The need to apply a major revision stems from the comments made by referee # 1 regarding the cloud model properties. Along the process of responding to the comments, it was realized that the cloud effective radius assumed in the C1 cloud model adopted in the present work may not be an optimal representation of the stratocumulus clouds observed in the Southeastern Atlantic Ocean during the ORACLES operation period.

The figure shown below displays the histogram of the cloud effective radius (CRE) data derived from the standard Aqua/MODIS MYD06 cloud product over the Southeastern Atlantic Ocean from August through October 2016 to 2018. The CRE histograms for the months of August to October reveal a distinct peak around 11-12 μm with a wide distribution ranging from 6 μm to 26 μm, albeit 70%-80% of the data points are found to be within 6-18 μm. Meyer et al. (2015) have shown that the presence of partially absorbing carbonaceous smoke aerosols over clouds in this region has only a marginal impact on the retrieved CRE.

[Figure]

*Figure 1 Histograms of the Aqua/MODIS cloud effective radius retrievals over the Southeastern Atlantic for the ORACLES period of August through October 2016-2018.*

The C1 cloud model adopted in our work assumes the liquid droplet size distribution that follows the modified Gamma distribution with α, β, and γ parameters of 6.0, 1.5, and 1.0, respectively. The CRE value, when calculated using these parameters, was 6 μm, which is an underestimation compared to the MODIS CRE histograms discussed above.

Based on the MODIS CRE histograms corresponding to the region and period of interest to our study, therefore, we have revised the cloud model with an optimal CRE value of 12 μm. This is achieved by changing the α parameter from 6.0 to 15.0 and keeping the β and γ parameters unaltered.

**The entire analysis presented in the originally submitted manuscript was reperformed with the revised cloud model that required recreating the aerosol-cloud LUTs for the above-cloud SSA retrieval and its sensitivity analysis. The following are the major changes we noticed in the results.**

1. The derived above-cloud SSAs are now lower in magnitudes by up to 0.01 and 0.02 at the near-UV and VIS-NIR wavelengths, respectively.
2. Most importantly, the comparison of the satellite-retrieved SSA against ORACLES in situ measurements, 4STAR inversions, and ground-based AERONET inversions improved noticeably, where the overestimation seen in the satellite retrievals is now either eliminated or largely reduced.
3. The results of the sensitivity analysis tabulated in Tables 3 and 4 show marginal change without its interpretation discussed in the original version of the manuscript.
4. The revised paper now also includes the uncertainty estimates in the retrieved aerosol-corrected COD resulting from perturbing the assumed aerosol and cloud parameters. These results are tabulated in Tables 5 and 6.

**All relevant figures and tables are updated with the revised calculations and results.**

Referee comment **RC in RED**
Author's response **AR in BLACK**

**General comments from Referee # 2**

"Retrieving UV-VIS Spectral Single-scattering Albedo of Absorbing Aerosols above Clouds from Synergy of ORACLES Airborne and A-train Sensors" by Hiren Jethva et al.

This paper introduces a synergy algorithm to improve the retrieval of SSA of aerosols over clouds, which is dearly needed to improve estimates of aerosol direct effects and aerosol forcings. The method is based on a 'color ratio' approach which relies on the spectral absorption of aerosols in the UV and visible, which is true mostly for smoke. Using two spectral measurements, two pieces of information can be retrieved, like AOD and COD when assuming the spectral properties of the aerosols. In the current paper, a method is proposed to instead constrain the AOD above the cloud and retrieve the COD and SSA of the aerosols, which can be done if independent above-cloud AOD is available, in this case using airborne measurements from three ORACLES campaigns over the tropical Atlantic. This area is interesting for the smoke over persisting stratocumulus, providing an obvious focus for the application provided analyses.

The presented approach is interesting since SSA retrievals, especially over clouds, are scarce. The application of the ORACLES aircraft campaign data is an interesting approach to test the satellite retrievals with a proper constraint on the aerosol properties, which is almost always missing. However, there are some major issues related to the presentation of the work.

First, the introduction is quite extensive, but it reads like an advertisement for the color ratio approach, while many pioneering work has been neglected.

Some examples:

l.55-56: retrieval of elevated aerosol from satellite: add Hsu, et al (2003)
Hsu et al. (2003) did not perform actual retrievals of aerosol-cloud properties of the elevated smoke layer above the cloud deck. Instead, they showed the impact of partially absorbing smoke layer overlying clouds on the satellite measurements of Reflectivity, spectral reflectance, and reflected solar radiation from clouds. Since Hsu et al. (2003) paper was about the direct effects of aerosols above clouds, we have added the reference following the statement, "Unlike the cloud-free conditions in which aerosols generally produce a cooling effect on climate, the presence of elevated layers of absorbing aerosols over clouds can potentially exert a large positive forcing through enhanced atmospheric heating resulting from aerosol-cloud radiative interactions (Hsu et al., 2003; Keil and Haywood, 2003; Chand et al., 2009; de Graaf et al., 2012; Zhang et al., 2016; Kacenelenbogen et al., 2019)".

l 77. Satellite observations have unambiguously shown the presence of absorbing aerosols above clouds over several regions of the world on a monthly to seasonal scale: add Peers, et al (2019).
Peers et al. (2019) demonstrates the retrieval capability of aerosols above clouds (AOD, COD, and CRE) from MSG/SEVIRI. Therefore, we've mentioned this reference in the statement "These techniques have shown the potential to retrieve ACAOD using measurements from different A-train sensors, including CALIOP/CALIPSO (Hu et al., 2007; Chand et al., 2008), Aura/OMI (Torres et al., 2012), Aqua/MODIS (Jethva et al., 2013; Meyer et al., 2015), Parasol/POLDER (Waquet et al., 2009), and MSG/SEVIRI (Peers et al., 2019)."

l. 533. The magnitudes of bias in the apparent COD depend on the strength of aerosol absorption and backscattering as well as on the actual value of COD (Jethva et al., 2018). The proper reference is: Haywood, et al. (2004).
Meyer et al. (2015) also showed impact of absorbing aerosols above clouds on the retrieved COD and CRE. We've added both Haywood, et al. (2004) and Meyer et al. (2015) papers here.

Figure 1 and l.133-147. The physical basis has been explained in the original papers on the color ratio (as amply referenced in the introduction) and are rather illegibly repeated in Figure 1 (the six panels are difficult to read).
The physical basis of the color ratio method explained in our previous papers corresponds to the simultaneous retrievals of ACAOD and COD, by assuming a fixed aerosol model, including SSA. In this paper, since we're proposing to retrieve SSA above clouds by constraining the AOD as an input, a separate figure demonstrating the sensitivity of TOA reflectance and ratio is needed. The six-panel Figure 1 shows the distinct sensitivity of color ratio and reflectance to the varying imaginary part of the refractive index or SSA (since size distribution is fixed) and COD for three different values of input ACAOD. Therefore, we will retain Figure 1 in its original form.

l. 122. "this kind of situation produces a strong 'color ratio' effect, which can be seen in the TOA measurements made by satellite sensors such as OMI (Torres et al., 2012) and MODIS (Jethva et al., 2013).": Actually shown in TOA reflectances from OMI and MODIS combined in De Graaf, et al. (2019).
Agreed. The sentence is modified as "In other words, this kind of situation produces a strong 'color ratio' effect, which can be seen in the TOA measurements made by satellite sensors such as OMI (Torres et al., 2012), MODIS (Jethva et al., 2013), and OMI-MODIS combined (De Graaf, et al., 2019).

Some more examples referencing own work

l 530. "The lofted layers of absorbing aerosols over the clouds attenuate light reflected by the cloud top through scattering and absorption. This effect reduces cloud-reflected upwelling UV (Torres et al., 2012) and VIS-NIR radiation (Jethva et al., 2013; Meyer et al., 2015),"

Under the pretence of "benchmarking" ones own RTM:

l. 366 The model results have been benchmarked against the results published in the literature, as well as used in the operational ACAOD product of OMI (Jethva et al., 2018), research-level ACAOD retrievals from MODIS (Jethva et al., 2013, 2016), DSCOVR-EPIC (Ahn et al., 2021), and S5p-TROPOMI (Torres et al., 2020).

Here, we want to say that the VLIDORT model results were compared against the RT simulations from benchmark RT codes, such as TOMRAD and the ARIZONA code used internally within the trace gas-aerosol groups in NASA Goddard. The same VLIDORT code was further upgraded for the joint aerosol-cloud simulations and used to create look-up tables for the above-cloud AOD retrieval from OMI, EPIC, and TROPOMI. The validation of the ACAOD product from these sensors against the ORACLES airborne direct measurements revealed a good level of agreement, suggesting the accuracy and reliability of the VLIDORT RT code.

We have rephrased this sentence for better clarity.

"The VLIDORT model results were compared against the published literature and the RT simulations from benchmark in-house RT codes used within the trace gases and aerosol groups at NASA Goddard. The VLIDORT code was further upgraded for the joint aerosol-cloud simulations and used to create look-up tables for the above-cloud AOD retrieval from Aura/OMI, DSCVOR-EPIC, and S5p/TROPOMI sensors. The validation of the ACAOD product from these sensors against the ORACLES airborne direct measurements revealed a satisfactory agreement within the expected uncertainties (Jethva et al., 2018; Ahn et al., 2021; Torres et al., 2020), thereby establishing the reliability of the VLIDORT RT code in the present application."

Repetition of references:

l. 85 These techniques have shown the potential to retrieve ACAOD using measurements from different A-train sensors, including (...), Aura/OMI (Torres et al., 2012), Aqua/MODIS (Jethva et al., 2013; ....),

l 102. This work builds upon the 'color ratio' (CR) technique previously applied to OMI (Torres et al., 2012; Jethva et al., 2018) and MODIS (Jethva et al., 2013, 2016) sensors.

We don't see any problem with this sentence or self-citation, as it is true that the technique proposed in the current paper is an extension of the CR algorithm that our group has developed and applied previously to the OMI and MODIS observations for retrieving ACAOD.

l. 122. In other words, this kind of situation produces a strong 'color ratio' effect, which can be seen in the TOA measurements made by satellite sensors such as OMI (Torres et al., 2012) and MODIS (Jethva et al., 2013).

Addressed in the one of the previous comments.

l. 364. We employ the VLIDORT code (Spurr, 2006) (...) - This should be merged with the info on l. 130.
We retain this sentence at its original place.

l. 718 - l.724: For instance, Jethva et al. (2018), through theoretical simulations, quantified (...) - Remove from the summary or merge with results in 5.4.

The sentence is now shortened and rephrased as "A perturbation of ±0.03 in the assumed aerosol SSA can lead to the uncertainty in the retrieved ACAOD in the range -20% to +50% at the UV and VIS wavelengths, the variation of which depends on the magnitudes of both ACAOD and COD."

Second, the manuscript lacks a comparison with existing work. Section 6 is interesting, if it would be compared to what we already know. This could easily be a simulation of the change in retrieved COD as a function of the aerosol absorption (in whatever form, SSA or AEE), which should not be too difficult with the RTM and LUTS already available. Alternatively, and more interesting would be to compare the actual measurements of the COD retrievals with and without ORACLES constraint aerosol above the cloud to other estimates of the change in COD, like e.g. in Haywood, et al (2004) or Peers, et al (2019). Now we learn nothing new. The main conclusion of the section is this, which should be put in perspective:

**AR:** Figure 11 (a) precisely demonstrates the effect of accounting (or not) for the presence of absorbing aerosols in the cloud optical depth retrievals. The revised manuscript includes the following description on the comparison of COD retrieved with and without using ORACLES-constrained ACAODs.

"*The aerosol-corrected CODs are found to be higher compared to apparent CODs (not corrected for aerosols) by an average (standard deviation) value of 26% (±20%) and 9% (±7%) at 388 nm and 860 nm, respectively. In other words, if not accounted for the presence of absorbing aerosols above the clouds, the retrieved COD turns out to be about -18% (±14%) and -8% (±6%) biased low compared to the aerosol-corrected CODs at UV and NIR wavelengths, respectively. The linear regression fits show positive slopes of 1.56 and 1.21 at 388 nm and 470 nm, respectively, with a negative intercept of ~1-1.5, which points to the inherent uncertainties involved in the proposed inversion technique at lower COD values.*"

Furthermore, a parameterization of the differences in CODs as a function of the above-cloud AAOD shown in Figure 11 (b) and (c) brings relatively new information that the former quantity is a bi-function of both the true COD underneath the aerosol layer and AAOD. These results are important for a more accurate estimation of the radiative effects of aerosols above clouds.

RC: l. 556 - 560. "These results are significant and further signify the importance of aerosol absorption above clouds in the UV to VIS-NIR spectral region in at least two ways. First, aerosol absorption above clouds, if not accounted for in the remote sensing inversion, can potentially introduce a negative bias in the retrieved cloud optical depth retrieval, whose magnitude depends on the strength of aerosol absorption (AAOD) and cloud brightness (COD) underneath the aerosol layer. " It would be nice to see how significant actually the results are, by comparing it with the expectations and results from earlier studies.

**AR:** The negative bias in the retrieved cloud optical depth, resulting from not accounting for the presence of overlaying aerosol layers, has been shown earlier in our previous papers and by other researchers. The results presented in this paper, i.e., Figure 11 (a), are a reaffirmation of the previous findings, using the airborne-satellite synergy. The impact of these results on the estimation of aerosol direct radiative effects requires a separate study, which is out of the scope of the presented work.

RC: Also, the sensitivity section is interesting and the amount of work is appreciated, it provides the necessary feel of the accuracy of the proposed method, and AMT is the right place for this kind of information. However, the provided tables of increased or decreased numbers of quantities are tediously repeated from the table into the text, and not compared to anything. It would quite interesting to have a few sensitivity studies from the RTM be compared to the retrievals. Then we

could actually see if what is expected is also being found from the satellite measurements when aerosol properties are properly and sufficiently constrained from the aircraft campaigns, or that still more information is needed.

**AR:** The approach we adopted to quantify the theoretical uncertainty presented in our paper combined both the radiative transfer simulations or aerosol-cloud LUT and the perturbation of actual measurements of ACAOD, ALH, AAE, and PSD. For instance, perturbing ACAOD and ALH did not require the generation of additional LUT. Instead, both quantities varied within the expected range of uncertainties (due to errors in measurements and/or airborne-satellite collocation) to calculate the change in the retrieved SSA. On the other hand, the sensitivity of the SSA retrievals to the change in AAE, PSD, $NO_2$, and cloud effective radius did require additional RTM and LUTs. Furthermore, the entire sensitivity analysis was applied to the actual airborne-satellite dataset of HSRL-2-OMI-MODIS synergy for the ORACLES August-September 2016 to quantify the realistic error matrix in the retrieved above-cloud SSA and cloud optical depth (the latter is added in the revision).

We don't think sensitivity studies purely based on RTM would add any additional information or improve the error matrix reported in our paper.

RC: Last, the manuscript lacks a proper motivation of the proposed synergy method. Obviously, the ORACLES data can only be used to test a proper constraint of the aerosol during the flights. CALIOP is mentioned as a replacement of the ACAOD measurements, noticing that (l776.) "CALOP [sic], OMI, and MODIS sensors fly in formation and make measurements within a few minutes of time difference.", in the present tense. The Calipso mission has ended and OMI is hardly producing useful measurements any more. Is the purpose to derive a SSA-over-cloud climatology from the past 10-15 years? Or can this also be applied to existing and future missions?

AR: We believe the motivation for developing the proposed synergy method is adequately discussed in the Introduction section. The above-cloud AOD retrieved from the passive sensors (such as OMI & MODIS) using multispectral algorithms is sensitive to the choice of aerosol model, as shown in our previous publications as well as by others (e.g., Meyer et al., 2015). Of various assumptions made in the model, the single-scattering albedo, followed by the spectral dependence of absorption (or AAE), is known to be the largest source of uncertainty in the retrieved ACAOD. These uncertainties, if not addressed, can proportionately result in errors in above-cloud aerosol radiative effects. Radiative transfer simulations show that the top-of-atmosphere measurements are sensitive to columnar ACAOD, SSA, and COD. Therefore, constraining the ACAOD in the algorithm allows two-parameter retrievals of SSA and COD. To test our hypothesis and the proposed algorithm, we used direct airborne measurements of ACAOD collected during the ORACLES campaign synergized with the OMI and MODIS observations over the Southeastern Atlantic Ocean.

A successful application of the synergy algorithm to the ORACLES and OMI-MODIS observations opens a new opportunity to extend the algorithm to CALIOP-OMI-MODIS synergy. Yes, we acknowledge that the CALIPSO mission has ended, and its light was turned off sometime this summer. OMI on Aura is also heading towards the end of its life. So, the intended application

of the proposed algorithm is to produce the past record of SSA above clouds from the coincident and overlapped period, i.e., 2006-2020.

RC: Or can this also be applied to existing and future missions?
AR: The proposed algorithm relies on accurate measurements of ACAOD for the retrieval of above-cloud aerosol SSA. The ACAOD algorithms applied to the existing passive sensors, such as OMI, MODIS, and POLDER, are dependent on the optical-microphysical models of aerosols. These retrievals carry errors due to the uncertainties in the assumed aerosol model. We're unaware of any existing satellite sensor that provides direct measurements of AOD above the cloud. Regarding the future missions, an HSLR-2 like active lidar sensor onboard satellite mission would provide the desired direct measurements of ACAOD for the proposed SSA retrievals.

I feel there is an increased interest in aerosol-cloud-radiation interaction and upcoming mission focus exactly on this topic, like 3MI/Sentinel-5 and EarthCare. The results presented here would be interesting for those missions as well and deserve a discussion, especially in a section named FUTURE IMPLICATIONS.
We have added a brief discussion on the applicability of the proposed synergy method to the future AOS and EarthCARE missions in section 8.1: Application of the Proposed Synergy Method to Existing and Future Missions.

Some textual issues are listed below and should be addressed, but more importantly some terms and phrases must be revisited:

(1)

l 639. "A strong relative spectral dependence of 20% in the imaginary part of the refractive index between 354 nm and 388 nm, resulting in AAE of 2.45-2.60 "

This should be rephrased in proper physical terms. It is unclear what is meant, but from the context one can expect that the spectral variation of the refractive index is meant, which cannot be "strong". It also does not result in AAE of 2.45, it is expressed in AAE.
We have reworded the sentence for clarity: "*The smoke particles are assumed to be, so-called 'colored aerosols', representing spectrally varying imaginary part of the refractive index, in which the latter is assumed to be 20% higher at 354 nm than that at 388 nm (see Table 2). This spectral dependence in the imaginary part of the refractive index translated into AAE in the range 2.45-2.60, which is expected to adequately represent the organic content of the carbonaceous smoke from biomass burning.*"

Similar arguments hold for:

l. 753. "reduced spectral dependence of absorption". Please, state the reduction in AEE (if this is meant).
Yes, we meant the reduced spectral absorption as reduction in AAE. The sentence is now rephrased as "*Changing the near-UV aerosol model with reduced spectral dependence of absorption or*

*relatively lower AAE brought the observed RMSD down from 0.05 to ~0.03—an improvement of 0.02 in SSA.*"

l. 757. "stronger absorption strength" Please, rephrase.
The sentence is now rephrased as *"The satellite retrievals, ORACLES in situ measurements, and ground-based inversion from AERONET all show increased absorption (lower SSA) in August, relatively weaker absorption in September, and even lower absorption (higher SSA) in October"*.

(2)
Replace all instances of 'spacetime' with spatiotemporal (or something better).
Both 'spacetime' and 'spatiotemporal' collocation convey the same meaning. However, we have adopted reviewer's suggestion to use 'spatiotemporal' in the revision.

**Minor issues:**
l. 64-67: A better (or additional) reference of the critical cloud fraction is Chand, et al (2009).
Chand et al. (2009) varied ACAOD, Angstrom Exponent, and asymmetry parameter, in their RT simulations, but kept SSA fixed as 0.85±0.02, whereas Eswaran et al. (2015) showed the sensitivity of CCF to varying SSA. We find both studies complementary to each other. We have added the following sentence referring to the work of Chand et al. (2009).

"*Along the same line, Chand et al. (2009) estimated the critical cloud fraction to be 0.4 for the Southeastern Atlantic Ocean, and further stated that it is strongly sensitive to the amount of solar radiation absorbed by aerosols, which in turn depends on the columnar aerosol loading and absorption capacity (SSA) of aerosols, and the albedo of the underlying clouds.*"

l. 181-184. This information seems out of place for the current paper.
The information given here simply states that the 4STAR sun photometer was successfully operated during the past field campaigns and provided invaluable optical measurements of aerosols and trace gases.

Section 5.1 should be merged with section 3.1.1. and repetition of information removed.
Sections 3.1.1, 3.1.2, and 3.1.3 briefly describe the ORACLES airborne sensors and their respective datasets, whereas section 5.1 presented the actual results of ACAOD from HSRL-2 and 4STAR, that are used in the above-cloud SSA inversion. We want to keep this distinction between the dataset description and their results.

l. 493: remove "noted"
Suggestion accepted.

l. 507. RMSD is not defined.
RMSD is now defined as the root-mean-square-difference.

The AERONET station Mongu is repeatedly mentioned and explained in the text. Remove all repetition.

We've removed the repletion as much as we could.

l. 607. Start a new paragraph here.
Suggestion accepted.

l. 630 lesser errors -> smaller errors.
Accepted.

l. 669 define PSD.
PSD is defined as the particle size distribution.

l. 730-735: Merge with the methods section (or simply remove this repetition).
We realize that the information provided here repeats what is already described earlier in the paper. So, we have accepted the reviewer's suggestion and removed line # 730-735 in the revision.

l. 774. one reason -> One reason of the bias
This should be line 744. The sentence corrected as "One reason for the bias, among several others…"

l. 776 CALOP -> CALIOP
Corrected.

References

de Graaf, M., Tilstra, L. G., and Stammes, P.: Aerosol direct radiative effect over clouds from a synergy of Ozone Monitoring Instrument (OMI) and Moderate Resolution Imaging Spectroradiometer (MODIS) reflectances, Atmos. Meas. Tech., 12, 5119–5135, https://doi.org/10.5194/amt-12-5119-2019, 2019.

Haywood, J. M., Osborne, S. R. and Abel, S. J. (2004) The effect of overlying absorbing aerosol layers on remote sensing retrievals of cloud effective radius and cloud optical depth. Quarterly Journal of the Royal Meteorological Society, 130 (598). pp. 779-800. ISSN 1477-870X

Hsu, N. C., Herman, J. R., and Tsay, S.-C. (2003), Radiative impacts from biomass burning in the presence of clouds during boreal spring in southeast Asia, Geophys. Res. Lett., 30, 1224, doi:10.1029/2002GL016485, 5.

Waquet, F., Riedi, J., Labonnote, L. C., Goloub, P., Cairns, B., Deuzé, J., and Tanré, D.: Aerosol Remote Sensing over Clouds Using A-Train Observations, J. Atmos. Sci., 66, 2468-2480, https://doi.org/10.1175/2009JAS3026.1, 2009.

Peers, F., Francis, P., Fox, C., Abel, S. J., Szpek, K., Cotterell, M. I., Davies, N. W., Langridge, J. M., Meyer, K. G., Platnick, S. E., and Haywood, J. M.: Observation of absorbing aerosols above clouds over

the south-east Atlantic Ocean from the geostationary satellite SEVIRI – Part 1: Method description and sensitivity, Atmos. Chem. Phys., 19, 9595–9611, https://doi.org/10.5194/acp-19-9595-2019, 2019.

---

## Author Response (AR2)

**egusphere-2023-1717**

**Retrieving UV-VIS Spectral Single-scattering Albedo of Absorbing Aerosols above Clouds from Synergy of ORACLES Airborne and A-train Sensors**

**Jethva et al.**

**Review round # 2**

**Response to the Anonymous Referee # 1**

We thank and appreciate the anonymous reviewer for taking the time to review our revised manuscript and offering constructive and valuable comments and suggestions.

The following are the one-to-one responses to a couple of comments made by referee # 1.

Referee comment **RC in RED**
Author's response **AR in BLACK**

**RC**: In Figure 9, when comparing 9a and 9b using different aerosol models, why does the total number of collocated data points change? It seems Figure 9b missed the data at Sep. 12, and it would affect the statistics. Is there any criteria used to keep the quality or reliability of the retrieval? An apple-to-apple comparison is required to give valid conclusions.

**AR:** The total number of collocated data points changed from N=23 (Figure 9 a) to N=20 (Figure 9 b) due to the change of the aerosol model, in which the latter aerosol model could not retrieve SSA at 388 nm due to an out-of-bound issue. The revised aerosol model with the relative spectral dependence between the two near-UV wavelengths could not resolve the observations within the aerosol-cloud look-up table. No criteria adopted in retrieving the data shown in Figure 9, where retrieved values for all AODs and CODs are included.

To make both datasets/plots equivalent in terms of the number of matchups, we have excluded the three additional collocated data points from Figure 9 (a) and included a revised figure (along with statistics) in the manuscript as well as shown in this report.

**RC:**  The additional Tables 3-6 provided important and helpful information, but they look cumbersome and I suggest the authors to make them more concise.

**AR:** We have considered the referee's suggestion and combined Tables 3 and 4, and Tables 5 and 6, showing the error matrix of the retrieved SSA at 388 nm (OMI) and 470 nm (MODIS) in the same table. The revised format looks modular and facilitates a direct comparison of errors at 388 nm and 470 nm to various sources of uncertainties in the algorithmic input parameters.

[Figure]

**Figure 9.** Comparison of spatiotemporally collocated above-cloud aerosol SSA retrieved from OMI (388 nm) against those derived from 4STAR sunphotometer sky scan observations for the ORACLES-1 September 2016 campaign. The OMI SSA retrievals using the original aerosol model (a, left) with 20% relative spectral dependence in the imaginary part of the refractive index (AAE ~2.45-2.60) and modified aerosol model (b, right) with 10% spectral dependence in the imaginary index (AAE ~1.72-1.87) are evaluated against those of 4STAR inversions. The Q_0.03 and Q_0.05 are the % matchups falling within the relative difference of ±0.03 and ±0.05, respectively. The OMI-4STAR matchups are color-coded according to the date of observations shown in legends. The sizing of the circles corresponds to the magnitude of coincident AOD (500 nm). The statistical measures of the comparison are printed in the lower left of each plot.

egusphere-2023-1717

Retrieving UV-VIS Spectral Single-scattering Albedo of Absorbing Aerosols above Clouds from Synergy of ORACLES Airborne and A-train Sensors

Jethva et al.

Review round # 2

**Response to the Anonymous Referee # 2**

Referee comment **RC in RED**
Author's response **AR in BLACK**

RC: The major changes requested in the first review are clearly not accepted by the authors. The response extensively explains why the original criticism should be ignored, instead of changing the manuscript. Of course, the authors know their work best and this may be the best way forward. However, the new manuscript contains new material that requires a careful consideration. It would not make sense to do this review again, knowing that it will not be accepted by the authors. Therefore, I suggest to look for new reviewers.

**AC:** The referee's comments on the response and revised manuscript look unreasonable. We have put our sincere efforts to address and include each of referee's comments and suggestions in the revision. The major change introduced in the revised manuscript after the first round of review was the change in the cloud model effective radius, which was adequately justified and described in the response as well as in the manuscript.

In the following response, we further clarify each of the comments and concerns raised by the referee and its consideration in revising our manuscript.

1) RC: "Second, the manuscript lacks a comparison with existing work. Section 6 is interesting, if it would be compared to what we already know. … The main conclusion of the section is this, which should be put in perspective: … "These results are significant and further signify the importance of aerosol absorption above clouds in the UV to VIS-NIR spectral region in at least two ways. First, aerosol absorption above clouds, if not accounted for in the remote sensing inversion, can potentially introduce a negative bias in the retrieved cloud optical depth retrieval, whose magnitude depends on the strength of aerosol 560 absorption (AAOD) and cloud brightness (COD) underneath the aerosol layer. " It would be nice to see how significant actually the results are, by comparing it with the expectations and results from earlier studies."

I suggest that you include some more discussion in the paper, either in a separate discussion section or by adding a paragraph to the conclusion section. Relating your work to previous findings and giving an outlook on additional work needed in the future would certainly improve your paper (see also the next two comments).

**AC:** We understand that the reviewer's comments are associated with section 6 describing the cloud optical depth (COD) retrievals. Let us emphasize here that our paper focuses on demonstrating the new retrieval technique to retrieve spectral aerosol SSA from space when combined with the independent direct measurements of aerosol loading above clouds. Of course, the aerosol-corrected CODs are also co-retrieved in this method. The purpose of adding a section on COD results was two-fold:

1) to show the impact of aerosol absorption effects on COD retrievals, which, if ignored, can introduce a significant bias in the cloud retrievals, as shown by other researchers in previous studies, and

2) to further parameterize the bias in COD values as a function of aerosol absorption effects on CODs. The latter results have shown us that given the same magnitudes of aerosol absorption above clouds, the magnitudes of bias in CODs depend on the true value of COD underneath the aerosol layer.

These results have important implications not just in cloud remote sensing but also in correctly estimating the aerosol radiative effects over clouds. Both these research topics are currently out of the scope of the present analysis, but they should be further investigated in depth in future studies.

2) RC: "Also, the sensitivity section is interesting and the amount of work is appreciated, it provides the necessary feel of the accuracy of the proposed method, and AMT is the right place for this kind of information. However, the provided tables of increased or decreased numbers of quantities are tediously repeated from the table into the text, and not compared to anything. It would quite interesting to have a few sensitivity studies from the RTM be compared to the retrievals. Then we could actually see if what is expected is also being found from the satellite measurements when aerosol properties are properly and sufficiently constrained from the aircraft campaigns, or that still more information is needed."

Here, it would probably be helpful to add some explanation at the beginning of Sec. 7 to motivate your approach. Even without performing additional sensitivity studies, some further discussion might be helpful (see my suggestion above).

**AC:** The sensitivity analysis entirely based on RTM simulations is a standard approach for quantifying the theoretical uncertainties in satellite-based inversions. On the other hand, quantifying the actual uncertainties involved in various assumptions made in the algorithm is a daunting task since the information on the true state of aerosol and cloud parameters is often unavailable. Therefore, we adopted another approach in which the RTM simulations, or aerosol-cloud look-up tables in the present context, were first carried out by considering uncertainties in different algorithmic assumptions individually. In the next step, the revised aerosol-cloud look-up tables were used in the inversion algorithm and applied to the actual airborne-satellite measurements. The resultant changes in the retrievals were then interpreted as the expected uncertainties in the retrieved spectral above-cloud aerosol SSA caused by realistic uncertainty in each algorithmic assumption separately. The suggested approach integrates different sets of RTM simulations and actual observations to derive realistic estimates of the errors in the derived aerosol SSA retrievals.

We have added the above description at the beginning of Section 7 in the revised paper.

Regarding the comment about "not compared to anything", we assume that the comment is related to the uncertainty estimates reported in the paper. Since the proposed retrievals of aerosol SSA above the clouds and associated uncertainty estimates are first-of-its-kind information from the satellite measurements, there aren't well established equivalent datasets available to facilitate a direct comparison. However, we already have made a reasonable effort to compare our satellite-based above-cloud aerosol SSA retrievals with the *in situ* measurements, airborne 4STAR sunphotometer-based SSA inversions made during the ORACLES campaign, as well as ground-based AERONET SSA datasets (Figures 8, 9, and 10). Overall, we find the satellite-based retrievals of SSA are found to compare well with these independent sets of measurements on an average sense, albeit all these SSA datasets do show some variability, which can be partly attributed to the inherent uncertainties involved in different techniques of measurements and inversion procedures.

**3) RC:** "Last, the manuscript lacks a proper motivation of the proposed synergy method. Obviously, the ORACLES data can only be used to test a proper constraint of the aerosol during the flights. CALIOP is mentioned as a replacement of the ACAOD measurements, noticing that (l776.) "CALOP [sic], OMI, and MODIS sensors fly in formation and make measurements within a few minutes of time difference.", in the present tense. The Calipso mission has ended and OMI is hardly producing useful measurements any more. Is the purpose to derive a SSA-over-cloud climatology from the past 10-15 years? Or can this also be applied to existing and future missions?

We have adequately responded to the referee's comment during the first round of review. The response is added below for the reference.

AR: We believe the motivation for developing the proposed synergy method is adequately discussed in the Introduction section. The above-cloud AOD retrieved from the passive sensors (such as OMI & MODIS) using multispectral algorithms is sensitive to the choice of aerosol model, as shown in our previous publications as well as by others (e.g., Meyer et al., 2015). Of various assumptions made in the model, the single-scattering albedo, followed by the spectral dependence of absorption (or AAE), is known to be the largest source of uncertainty in the retrieved ACAOD. These uncertainties, if not addressed, can proportionately result in errors in above-cloud aerosol radiative effects. Radiative transfer simulations show that the top-of-atmosphere measurements are sensitive to columnar ACAOD, SSA, and COD. Therefore, constraining the ACAOD in the algorithm allows two-parameter retrievals of SSA and COD. To test our hypothesis and the proposed algorithm, we used direct airborne measurements of ACAOD collected during the ORACLES campaign synergized with the OMI and MODIS observations over the Southeastern Atlantic Ocean.

A successful application of the synergy algorithm to the ORACLES and OMI-MODIS observations opens a new opportunity to extend the algorithm to CALIOP-OMI-MODIS synergy. Yes, we acknowledge that the CALIPSO mission has ended, and its light was turned off sometime this summer. OMI on Aura is also heading towards the end of its life. So, the intended application of the proposed algorithm is to produce the past record of SSA above clouds from the coincident and overlapped period, i.e., 2006-2020.

RC: Or can this also be applied to existing and future missions?
AR: The proposed algorithm relies on accurate measurements of ACAOD for the retrieval of above-cloud aerosol SSA. The ACAOD algorithms applied to the existing passive sensors, such as OMI, MODIS, and POLDER, are dependent on the optical-microphysical models of aerosols. These retrievals carry errors due to the uncertainties in the assumed aerosol model. We're unaware of any existing satellite sensor that provides direct measurements of AOD above the cloud. Regarding the future missions, an HSLR-2 like active lidar sensor onboard satellite mission would provide the desired direct measurements of ACAOD for the proposed SSA retrievals.

We direct the reviewer and the Editor to refer to the restructured sections 8.1 and 8.2 in the revised paper that discusses the potential application of the proposed synergy to the existing long-term A-train satellite record (CALIOP, OMI, and MODIS), and future satellite missions, such as NASA's AOS and ESA's EarthCare.

RC: I feel there is an increased interest in aerosol-cloud-radiation interaction and upcoming mission focus exactly on this topic, like 3MI/Sentinel-5 and EarthCare. The results presented here would be interesting for those missions as well and deserve a discussion, especially in a section named FUTURE IMPLICATIONS.

You have provided a detailed response to this comment, but you have not clearly indicated what has changed in the manuscript as a result. Obviously, you have added some text, but the structure with the new subsection 8.1 at the end of the paper is a bit strange, and the discussion is not complete (given your response). Again, improving the text in terms of discussion, conclusions and outlook might help to round off the paper.

**AC:** Considering the referee's comments during the first round of review, we have already added a discussion on the implications of the present synergy algorithm and resultant above-cloud aerosol absorption retrievals on the existing and future missions in sections 8.1 and 8.2, respectively. ESA's 3MI/Sentinel-5 sensor mentioned by the referee is an imager and would not provide direct measurements of AOD, like that from space-based HSRL lidar and/or airborne sunphotometer/lidar, which is required as an input to the present synergy algorithm. Therefore, we didn't include it in the discussion on future implications.

However, we have already added a brief discussion on ESA's upcoming EarthCare mission (https://earth.esa.int/eogateway/missions/earthcare#) and NASA's futuristic AOS mission (https://aos.gsfc.nasa.gov/mission.htm) potentially relevant to the present work. Of course, future suborbital airborne campaigns with targeted measurements of aerosols and clouds in the same atmospheric column, such as demonstrated with ORACLES airborne data in the present work, would provide additional opportunities to test and apply the synergistic approach for characterizing the aerosol absorption in the cloudy atmosphere.

In the revision, we've further restructured the final section 8 as follows.

**8 Final Remarks**
      8.1 Application of the Proposed Synergy Method to Existing Long-term A-train Record
      8.2 Future Implications